# P2P: Automated Paper-to-Poster Generation and Fine-Grained Benchmark

**Tao Sun[1,2], Enhao Pan[1], Zhengkai Yang[1], Kaixin Sui[1], Jiajun Shi[2], Xianfu Cheng[2],**
**Tongliang Li[4], Wenhao Huang[1], Ge Zhang[1,2], Jian Yang,[\*] Zhoujun Li[3\*]**

[1] ByteDance, China    [2] M-A-P    [3] Shenzhen Intelligent Strong Technology Co.,Ltd.
[4] College of Computer Science, Beijing Information Science and Technology University
`{buaast,jiaya,lizj}@buaa.edu.cn`

## Abstract

Academic posters are vital for scholarly communication, yet their manual creation is time-consuming. However, automated academic poster generation faces significant challenges in preserving intricate scientific details and achieving effective visual-textual integration. Existing approaches often struggle with semantic richness, structural nuances, and lack standardized benchmarks for evaluating generated academic posters comprehensively. To address these limitations, we introduce **P2P**, the first flexible, LLM-based multi-agent framework that generates high-quality, HTML-rendered academic posters directly from research papers. P2P employs three specialized agents—for visual element processing, content generation, and final poster assembly—each integrated with dedicated checker modules to enable iterative refinement and ensure output quality. To foster advancements and rigorous evaluation in this domain, we argue that generated posters must be assessed from two complementary perspectives: **objective fidelity and subjective quality**. So we establish **P2PEval**, a comprehensive benchmark featuring 1738 checklist items and a dual evaluation methodology (Fine-Grained and Universal). Our Fine-Grained Evaluation uses human-annotated checklists to objectively measure the faithful preservation of verifiable content from the source paper. Concurrently, our Universal Evaluation captures subjective, holistic quality by training a model to align with human aesthetic preferences across key design principles. We evaluate a total of 35 models. To power these advancements, we also release **P2PINSTRUCT**, the first large-scale instruction dataset comprising over 30,000 high-quality examples tailored for the academic paper-to-poster generation task. Furthermore, our contributions aim to streamline research dissemination while offering a principled blueprint for evaluating complex, creative AI-generated artifacts. The code is on the https://github.com/multimodal-art-projection/P2P.

## 1 Introduction

Academic posters serve as a vital tool in scholarly communication, distilling complex research into visually accessible formats to foster knowledge dissemination. However, manually creating these posters is a time-consuming and skill-intensive process, demanding a blend of content refinement and design proficiency. Automating academic poster generation thus offers a significant opportunity to streamline research dissemination, but it is a complex challenge that goes far beyond simple summarization or image placement.

The difficulty is twofold. First, the generation task itself requires a system to simultaneously master multiple sub-tasks (Qiang et al., 2016): (1) content distillation to identify and synthesize the most critical information; (2) visual textual integration to understand and contextually place figures and tables; and (3) structural reorganization to transform a linear document into a coherent, two-dimensional layout. Second, and equally challenging, is *evaluation*. A poster's quality is difficult to assess directly, as it requires balancing two distinct axes: **objective fidelity** and **subjective quality**.

---

[\*]Corresponding Authors

Objective fidelity, the faithful preservation of scientific claims and data, can be verified against the source paper. Conversely, subjective quality, which encompasses layout coherence and visual appeal, is inherently a matter of human aesthetic judgment. A robust evaluation must therefore address both (Que et al., 2024; Viswanathan et al., 2025).

Existing approaches to poster generation primarily rely on template-based or rule-driven methods (Xu & Wan, 2021), which often struggle to capture the semantic richness and structural nuances of academic documents (Qiang et al., 2019), typically decomposing the task into isolated subtasks like content extraction (Cheng et al., 2024b), panel attribute inference (Huang et al., 2022), and layout generation (Lin et al., 2024). Although recent advances in multimodal large language models (MLLMs) and large language models (LLMs) show promise in understanding document structures and visual-textual relationships (Jaisankar et al., 2024), their application to academic poster generation remains limited due to insufficient quality control mechanisms and the absence of standardized benchmarks for systematic evaluation.

To overcome these limitations and explore an underexplored research direction, we introduce a foundational ecosystem for automated academic poster generation, built on three synergistic contributions. The first is **P2P**, a novel and flexible multi-agent framework. It employs three specialized agents for visual processing, content generation, and layout orchestration, each integrated with a **checker reflection loop**. This iterative refinement mechanism mimics the human design process of drafting and revision, ensuring both scientific accuracy and structural integrity. The second contribution is **P2PINSTRUCT**, the first large-scale (30,000+ examples) instruction dataset for this underexplored domain, designed to train end-to-end models. The third is **P2PEVAL**, a comprehensive benchmark featuring 1738 checklist items, which implements our dual evaluation philosophy. **Fine-Grained Evaluation** measures objective fidelity by using LLM-as-a-Judge (Gu et al., 2024) to score generated posters against detailed, human-annotated checklists of verifiable content. While its **Universal Evaluation** captures subjective quality by training XGBoost (Chen & Guestrin, 2016) to emulate human annotators' holistic scores across 10 universal design and content criteria. We evaluate a total of 35 LLMs and MLLMs.

Our contributions are as follows:

- We propose **P2P**, the first multi-agent architecture for academic poster generation featuring an innovative checker-reflection mechanism for iterative refinement, providing a robust design pattern for complex document transformation tasks.
- We construct and release **P2PINSTRUCT**, the first large-scale (30K+) instruction dataset specifically designed to train models for the complete paper-to-poster generation workflow.
- We establish **P2PEVAL**, a new and comprehensive benchmark (1738 checklist items and 121 pairs) with a novel dual-evaluation framework that integrates human-annotated checklists and a predictive scoring model for robust, multifaceted analysis of poster quality.

## 2 METHODOLOGY

Our method, P2P, is a multi-agent framework that automates making posters. As a model-agnostic orchestration pipeline, it can be instantiated with various LLMs. It breaks down the complex job into small, manageable steps, with a specialized agent handling each one. A core innovation is the integration of a **checker-reflection mechanism** at each stage, enabling iterative refinement and ensuring high-quality output. The robustness of P2P also facilitates the creation of P2PINSTRUCT, a large-scale instruction dataset derived from the intermediate outputs of P2P.

### 2.1 P2P: MULTI-AGENT FOR PAPER-TO-POSTER GENERATION

As illustrated in Figure 1, the P2P workflow is orchestrated by three collaborative agents: the Figure Agent, the Section Agent, and the Orchestrate Agent. Each agent operates in conjunction with a dedicated checker module that triggers a reflection loop if its output fails to meet quality standards.

**Problem Formulation.** Given a research paper $D$ in a digital format (e.g., PDF), the task of P2P generation is to automatically synthesize an academic poster $P$ in a web-native format (HTML and CSS). We formalize this task as a sequential operation orchestrated by our specialized agents.

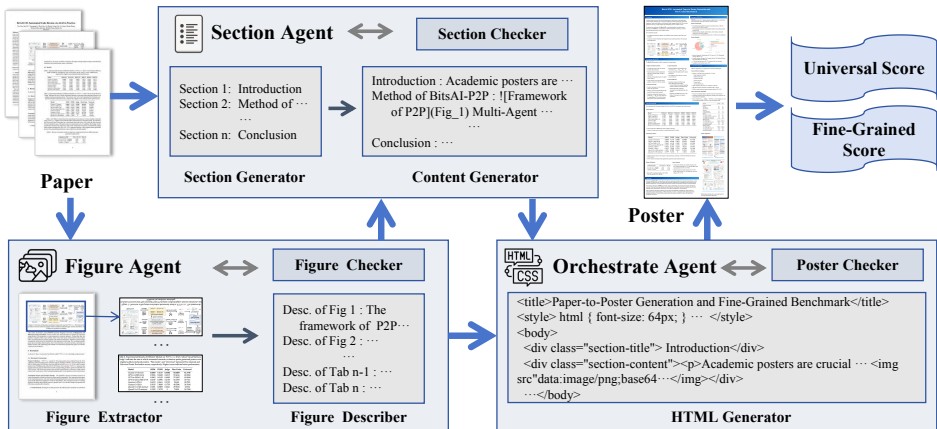

Figure 1: The multi-agent architecture of P2P: papers are processed by the Figure Agent for extraction and description of visual elements, the Section Agent for structural and content generation, and the Orchestrate Agent for poster assembly and HTML rendering. Each agent employs checker modules and reflection loops for iterative enhancement.

The overall process can be expressed as: $P = \mathcal{A}_{\text{Orch}}(\mathcal{A}_{\text{Sec}}(D, F), F)$, where $F = \mathcal{A}_{\text{Fig}}(D)$. Here, the process begins with the **Figure Agent**, $\mathcal{A}_{\text{Fig}}$, which processes the input paper $D$ to extract and describe its visual elements, producing an intermediate set of figures and tables $F$. Subsequently, the **Section Agent**, $\mathcal{A}_{\text{Sec}}$, generates the poster content $P_{\text{poster\_text}} = \mathcal{A}_{\text{Sec}}(D, F)$ by summarizing the paper $D$ while strategically referencing the visuals in $F$. Finally, the **Orchestrate Agent**, $\mathcal{A}_{\text{Orch}}$, assembles the textual content $P_{\text{poster\_text}}$ and visual elements $F$ into the final designed poster $P$.

**Figure Agent.** The Figure Agent is responsible for processing all visual elements within the input research paper. Its *Figure Extractor* component employs DocLayout-YOLO (Zhao et al., 2024), a state-of-the-art document layout detection model, to extract figures and tables. Concurrently, the *Figure Describer* identifies corresponding captions via spatial relation analysis. These components collaborate to synthesize semantic visual units by combining each extracted graphical component with its associated caption, yielding a set of described visual elements $\mathcal{F}_d = \{(v_1, c_1, \text{desc}_1), \dots, (v_n, c_n, \text{desc}_n)\}$. Here, $v_i$ denotes the raw visual element (the cropped image file and its metadata), $c_i$ its original caption, and $\text{desc}_i$ a detailed description generated by an MLLM, $M_{figure}$. The *Figure Checker* then validates this output by: (1) preventing duplicate extractions, (2) verifying the capture of all significant visual elements, and (3) confirming accurate visual-caption pairings. To ensure reliable pairings, an initial confidence threshold is applied to detected elements; this threshold is incrementally lowered if discrepancies arise between the counts of identified figures and captions, an iterative process repeated until sufficient alignment is achieved.

**Section Agent.** The Section Agent focuses on generating the textual content of the poster. Initially, the *Section Generator* analyses the input paper ($D$) to dynamically infer a detailed structural schema ($S$) for the target poster. This schema, represented as a JSON object, delineates crucial sections (e.g., Introduction, Methods, Results) and their intended content focus. Subsequently, the *Content Generator* synthesizes semantically coherent textual content for the poster, $P_{\text{poster\_text}}$, by utilizing the structural schema $S$, the original input paper $D$, and the detailed descriptions and indices of visual elements $\mathcal{F}_d$ provided by the Figure Agent. This textual content generation can be formally described as: $P_{\text{poster\_text}} = \mathcal{M}_{\text{text}}(D, S, \mathcal{F}_d)$, where $\mathcal{M}_{\text{text}}$ is a LLM specialized in text generation. $\mathcal{M}_{\text{text}}$ employs prompts not only to generate text but also to strategically integrate Markdown-style references to figure indices from $\mathcal{F}_d$ at optimal textual positions, ensuring contextual relevance and visual-textual alignment. The *Section Checker* scrutinizes the generated $P_{\text{poster\_text}}$ for: (1) coherence and logical flow, (2) completeness in covering core contributions, (3) faithfulness to the original paper's findings, and (4) correct and relevant referencing of visual elements. If inadequacies are detected, a reflection loop initiates a revision of the section structure or content by the Section Agent.

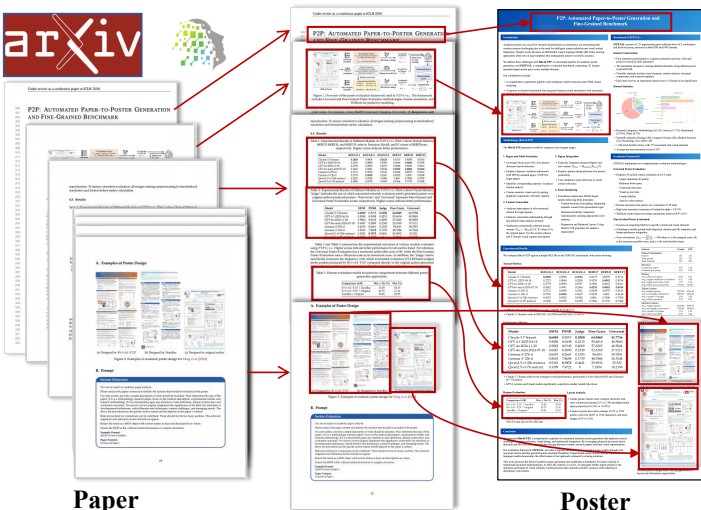

Figure 2: An example of the paper-to-poster transformation achieved by P2P, showing direct correspondences between elements in the input paper (left) and the generated academic poster (right).

**Orchestrate Agent.** The Orchestrate Agent integrates the visual and textual components into a cohesive and professionally formatted poster. The *HTML Generator* utilizes the Markdown-formatted text $P_{\text{poster\_text}}$ from the Section Agent and the actual visual elements (images/tables $\mathcal{F}_v$, where each figure is additionally provided with its width, height, and aspect ratio as supplementary information) extracted by the Figure Agent, to produce the poster in HTML and CSS. The Orchestrate Agent deliberately omits original captions from $\mathcal{F}_d$ in the final embedded visuals to improve visual clarity and maintain a concise academic presentation. The rendering process adheres to three principles: (1) Content-Structure Decoupling: Decouple semantics from presentation via modular CSS. (2) Institutional Identity Alignment: Customize color schemes to align with the logo of the institution or conference. (3) Responsive and Balanced Layout Generation: Use CSS flexbox for adaptive column structures and whitespace optimization. The *Poster Checker* evaluates the rendered poster for layout aesthetics and structural integrity, triggering iterative adjustments (via reflection) to resolve issues like unbalanced spacing or misaligned elements until the design meets professional standards.

Figure 2 illustrates the core transformation process facilitated by P2P. On the left, a multi-page academic research paper, sourced from repositories such as arXiv or conference proceedings like ICLR, serves as the input. On the right, the corresponding academic poster, generated by P2P, is displayed. The red arrows explicitly map key elements from the original paper, such as the title, specific figures, and sections, to their respective locations and representations in the final poster.

## 2.2 P2PINSTRUCT: A LARGE-SCALE INSTRUCTION DATASET

The P2PINSTRUCT dataset is derived from the P2P to support training of models for poster generation. Following P2P, we collect 30,460 high-quality instruction-response pairs spanning the complete poster generation workflow. For visual element processing, we prompt Claude to generate 16,848 figure-description pairs through the Figure Describer component, yielding descriptive texts averaging 192 tokens per visual element. For textual content generation, we collect 13,612 instruction-response pairs from the Section Generator, Content Generator, and HTML Generator components. These examples average over 3,300 tokens per response, demonstrating the complexity and richness of the generated content. A detailed account of its generation process, quality validation, and mitigation of potential biases is provided in Appendix D.

## 3 P2PEVAL: A FINE-GRAINED BENCHMARK FOR POSTER EVALUATION

As shown in Fig 3, we present a benchmark called P2PEVAL for evaluating academic posters.

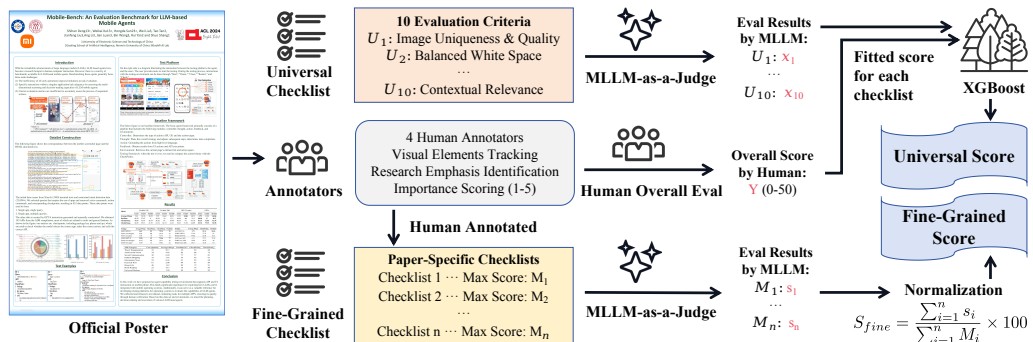

Figure 3: Overview of the poster evaluation framework used in P2PEVAL. P2PEVAL includes Universal and Fine-Grained Poster Evaluation, human annotators, and XGBoost for scoring.

## 3.1 BENCHMARK CONSTRUCTING

**Checklist Design.** The core of P2PEVAL lies in its detailed, human-curated and fine-grained checklists, designed to capture the essential elements of a poster. The annotation focuses on creating detailed checklists using official posters as ground truth. Our checklist design incorporates the following elements: **(1) Visual Elements**: Each figure or table present in the official poster constitutes an individual checklist item, evaluated based upon its presence and accurate representation. **(2) Content Analysis**: Each visual element is assessed regarding its textual consistency with the original poster and its visual prominence within the poster layout. **(3) Structural Components**: Annotators identify critical sections such as task definitions, experimental methodologies, and research conclusions within each poster panel. **(4) Research Emphasis**: Essential research findings, methodological details, and explicitly highlighted motivations (often noted by bold or prominent placement) form individual checklist items. **(5) Scoring System**: Each checklist item receives an importance-based score ranging from 1 to 5—minor details are rated as 1, core elements as 3, and critical components central to the paper as 5. The checklist items and their importance scores (1-5) are meticulously crafted by human experts with domain knowledge. Checklist format is listed in Appendix E.1.

**Annotation Protocol.** The annotation process was governed by a strict protocol to ensure quality and consistency. Each paper-poster pair is processed by four members of our annotation team. Three annotators independently create checklists, while a fourth, senior annotator serves as a verifier and integrator. This fourth annotator's role is to reconcile the three independent annotations by creating a superset (a union) of all identified checklist items. They also normalize the importance scores by calculating the average for each item across the initial annotations and rounding it to the nearest integer, ensuring a final, consensus-based score. Posters with major initial disagreements are excluded from the final test set to ensure data quality. For quality assurance in developing our checklists, we consult with researchers who have previously created posters (while maintaining anonymity). Their feedback confirms that our checklist design accurately reflects their poster creation priorities and decision-making process. Further details about our annotation team are provided in Appendix C.

**Dataset Collection and Statistics.** P2PEVAL consists of 121 paper-poster pairs collected from the ACL conference series (from 2022 to 2024) under CC4.0 license and from SciPostLayout (Wang et al., 2024a), which contains posters from F1000Research under the CC-BY license. For each pair, P2PEVAL preserves the original research paper in PDF format and the corresponding academic poster in both PDF and PNG formats. As shown in the Fig 4, P2PEVAL encompasses a broad range of research categories and disciplines. **The annotation process results in 1738 checklist items, with 775 associated with visual elements.** The scoring system yields an average per-item maximum score of 4.07. To ensure fairness and diversity, the benchmark includes papers from a wide range of scientific fields, as detailed in Appendix E.2. Our analysis in Appendix E.3 shows that model performance varies across topics, indicating the benchmark does not unfairly favor a single domain.

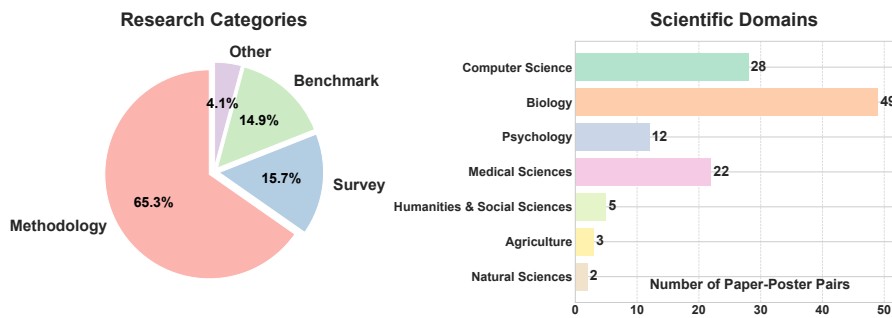

Figure 4: Distribution of P2PEVAL.

## 3.2 POSTER EVALUATION FRAMEWORK

Our evaluation pipeline consists of two complementary methodologies: Fine-Grained Poster Evaluation and Universal Poster Evaluation. We describe these two evaluation methods in detail below.

### 3.2.1 FINE-GRAINED POSTER EVALUATION

We design a Fine-Grained Poster Evaluation pipeline to measure a generated poster's fidelity to the key content selected by the original author. This metric focuses on the accurate representation of scientific information, not the imitation of a specific layout. The evaluation is a deterministic process that uses an LLM as an automated verification tool. The process consists of two steps:

**Step 1: Automated Verification.** For each item on the human-authored checklist (Section 3.1), we prompt an LLM (GPT-4o) to perform a strictly defined check and return relevant score: verify whether that specific, verifiable fact is present and accurately represented in the generated poster. The LLM's role is not subjective judgment. Each checklist item's maximum score is consensus-derived from multiple annotators, ranging from minor visual elements assigned a score of 1 to core research components scored at 5, reflecting their relative importance. This human-centred approach ensures that the scoring system inherently embodies human preferences and domain expertise.

**Step 2: Deterministic Scoring.** Then we programmatically aggregates these results from LLM. We formally define the final fine-grained evaluation score $S_{fine}$ as $S_{fine} = \frac{\sum_{i=1}^{n} s_i}{\sum_{i=1}^{n} M_i} \times 100$, where $S_{fine}$ is the normalized fine-grained evaluation score on a 0–100 scale, $s_i$ is the assigned score for the $i^{th}$ checklist item represented in the generated poster, $M_i$ denotes the corresponding maximum possible score for that item, and $n$ signifies the total number of checklist items. Consequently, the Fine-Grained Poster Evaluation score comprehensively assesses a generated poster's capability to faithfully preserve the original research's essential content and visual priorities. By emphasizing explicit fidelity to the original author's intended communication goals rather than generic quality alone, the approach enables clear comparative analyses across diverse poster generation methodologies.

### 3.2.2 UNIVERSAL POSTER EVALUATION

Universal Poster Evaluation employs a unified set of evaluation criteria, each evaluated independently on a discrete scale ranging from 0 to 5. These universal criteria ($U_1$ through $U_{10}$) include:

- $U_1$: Authorship and Title Accuracy
- $U_2$: Image Uniqueness and Quality
- $U_3$: Balanced White Space
- $U_4$: Contextual Relevance
- $U_5$: Optimal Visual-to-Text Ratio
- $U_6$: Dimension Appropriateness
- $U_7$: Visual Consistency
- $U_8$: Content Fidelity
- $U_9$: Information Flow Logic
- $U_{10}$: Self-Contained Explanation

While LLMs can score individual criteria, they often fail to replicate the complex, non-linear weightings humans apply when forming a holistic judgment. To address this, we developed a two-step

Table 1: Experimental results of different models on P2PEVAL. Higher scores indicate better.

| Model | Size | ROUGE-1 | ROUGE-2 | ROUGE-L | BERT[1] | Judge[2] | FineGrain[3] | Universal[4] |
|---|---|---|---|---|---|---|---|---|
| **Closed-Source Models** | | | | | | | | |
| Claude-3.7-Sonnet | 🔒 | 0.2745 | 0.0830 | 0.2527 | 0.8109 | 0.5537 | 65.3962 | **37.2474** |
| Claude-3.7-Sonnet[R] | 🔒 | 0.2734 | 0.0848 | 0.2516 | 0.8111 | **0.6281** | 65.8848 | 35.5062 |
| Claude-3.5-Sonnet | 🔒 | 0.2367 | 0.0615 | 0.2185 | 0.8081 | 0.2810 | 47.7385 | 30.2544 |
| GPT-4.1-2025-04-14 | 🔒 | 0.2459 | 0.0685 | 0.2281 | 0.8113 | 0.4793 | 60.2879 | 34.4700 |
| GPT-4.1-mini-2025-04-14 | 🔒 | 0.2616 | 0.0741 | 0.2407 | 0.8125 | 0.3388 | 55.3493 | 31.0697 |
| GPT-4.1-nano-2025-04-14 | 🔒 | 0.2169 | 0.0557 | 0.1990 | 0.8070 | 0.2066 | 41.3446 | 27.7149 |
| GPT-4o-2024-11-20 | 🔒 | 0.2395 | 0.0668 | 0.2217 | 0.8114 | 0.4959 | 55.4380 | 34.3888 |
| GPT-4o-mini-2024-07-18 | 🔒 | 0.2362 | 0.0732 | 0.2198 | 0.8167 | 0.2314 | 48.8879 | 30.8409 |
| OpenAI-o1[R] | 🔒 | 0.2385 | 0.0611 | 0.2200 | 0.8088 | 0.3103 | 56.8504 | 34.1659 |
| Seed1.5-VL[R] | 🔒 | 0.2160 | 0.0539 | 0.2026 | 0.8041 | 0.4050 | 62.4702 | 33.9840 |
| Seed-Thinking-v1.5[RT] | 🔒 | 0.2357 | 0.0701 | 0.2210 | 0.8113 | 0.4711 | 61.9632 | 34.6882 |
| Seed-Thinking-v1.5-m[R] | 🔒 | 0.2493 | 0.0767 | 0.2315 | 0.8116 | 0.3719 | 57.1457 | 33.2461 |
| Doubao-1.5-vision-pro | 🔒 | 0.2586 | 0.0849 | 0.2409 | 0.8089 | 0.0354 | 45.9282 | 14.0841 |
| YuanBao[5] | 🔒 | - | - | - | - | 0.0083 | 57.8677 | 31.5754 |
| **6B+ Models** | | | | | | | | |
| InternVL3 | 8B | 0.1980 | 0.0618 | 0.1847 | 0.7994 | 0.0776 | 33.3900 | 22.2245 |
| Qwen3[T] | 8B | 0.2563 | 0.0859 | 0.2373 | 0.8152 | 0.1835 | 45.0272 | 28.8107 |
| Qwen3[RT] | 8B | 0.2231 | 0.0619 | 0.2082 | 0.8125 | 0.2545 | 53.6611 | 32.4912 |
| Qwen2.5-VL | 7B | 0.1090 | 0.0414 | 0.1020 | 0.7645 | 0.0083 | 13.7417 | 13.0597 |
| **12B+ Models** | | | | | | | | |
| Gemma-3 | 12B | 0.2411 | 0.0764 | 0.2250 | 0.8096 | 0.0940 | 46.7903 | 27.3686 |
| InternVL3 | 14B | 0.2437 | 0.0736 | 0.2253 | 0.8132 | 0.0756 | 45.5513 | 25.6062 |
| **27B+ Models** | | | | | | | | |
| Gemma-3 | 27B | 0.2500 | 0.0794 | 0.2346 | 0.8133 | 0.2857 | 50.8931 | 28.7410 |
| Gemma-3[T] | 27B | 0.2536 | 0.0853 | 0.2372 | 0.8132 | 0.2417 | 52.1716 | 28.5901 |
| InternVL3 | 38B | 0.2440 | 0.0756 | 0.2258 | 0.8143 | 0.2333 | 52.6634 | 29.5850 |
| Qwen3[RT] | 3/30B | 0.2270 | 0.0637 | 0.2125 | 0.8120 | 0.2562 | 52.2125 | 31.1930 |
| Qwen3[RT] | 32B | 0.2314 | 0.0659 | 0.2168 | 0.8090 | 0.1736 | 46.0383 | 28.9479 |
| Qwen2.5-Coder[T] | 32B | 0.2666 | 0.0949 | 0.2487 | 0.8167 | 0.3884 | 55.9441 | 32.7935 |
| **72B+ Models** | | | | | | | | |
| Deepseek-R1[RT] | 37/671B | 0.1927 | 0.0461 | 0.1795 | 0.8015 | 0.5333 | 62.5013 | 33.9701 |
| Deepseek-V3[T] | 37/671B | 0.2371 | 0.0739 | 0.2232 | 0.8124 | 0.5041 | 59.6805 | 33.6045 |
| InternVL3 | 78B | 0.2424 | 0.0789 | 0.2245 | 0.8152 | 0.2773 | 51.2962 | 28.9230 |
| Qwen3[RT] | 22/235B | 0.2278 | 0.0625 | 0.2141 | 0.8077 | 0.3967 | 53.7927 | 31.4551 |
| Qwen2.5-VL | 72B | 0.2577 | 0.0909 | 0.2400 | 0.8148 | 0.2833 | 55.7929 | 32.3105 |
| Llama-4-Scout | 17/109B | 0.2806 | **0.1208** | 0.2625 | **0.8172** | 0.0413 | 35.7872 | 22.9738 |
| Qwen3-P2P[T6] | 8B | **0.2882** | 0.0955 | **0.2675** | 0.8135 | 0.4587 | 57.6622 | 32.4996 |
| Qwen2.5-VL-P2P[7] | 7B | 0.1939 | 0.0609 | 0.1797 | 0.7926 | 0.3140 | 37.3078 | 25.0337 |
| InternVL3-P2P[8] | 8B | 0.2744 | 0.0883 | 0.2551 | 0.8117 | 0.3772 | 51.9670 | 31.6206 |

[R] Reasoning/Thinking Mode. [T] Because they are text-only LLMs, we use `Claude-3.7-Sonnet` as the provider of Figure Describer. [1] F1 scores of BERTScore. [2] The rate at which LLM-as-a-Judger prefer generated posters over original author-produced posters. [3] Scores of Fine-Grained Poster Evaluation. [4] Scores of Universal Poster Evaluation. [5] Posters generated by Tencent's AI application called YuanBao. [6-8] Our model, built on `Qwen3-8B`, `Qwen2.5-VL-8B` or `InternVL3-8B`, is fine-tuned using P2PINSTRUCT respectively.

hybrid approach. First, we use an LLM to generate scores for the 10 discrete criteria. Second, we train an XGBoost model to predict a final holistic score from these 10 features, using 1,701 human ratings as ground truth. This hybrid methodology was a principled design choice, as it combines the LLM's strength in stable feature extraction with XGBoost's proven ability to learn complex, non-linear human preference functions. Specifically, the XGBoost model undergoes training with 10-fold cross-validation and utilizes 200 trees. The resulting predictive model exhibits strong performance, achieving an $R^2$ of 0.92, thus validating the reliability and effectiveness of our Universal Poster Evaluation pipeline. More details about Universal Poster Evaluation can be found in Appendix F.

Table 2: Results of pairwise human preference evaluations.

| Comparison (A vs. B) | Preferred or Tied (%) | Preferred (%) |
|---|---|---|
| P2P / YuanBao | 83.05 | 54.35 |
| P2P / Original | 57.63 | 35.59 |
| YuanBao / Original | 20.34 | 12.40 |

Table 3: Performance comparison of P2P across different output format.

| Output | FineGrain | Universal |
|---|---|---|
| HTML | **65.3962** | **37.2474** |
| SVG | 52.7408 | 30.6648 |
| LaTex | 56.8756 | 25.2585 |

## 4 EXPERIMENTS AND ANALYSIS

### 4.1 EXPERIMENTAL SETUP

We conduct comprehensive experiments to evaluate P2P against several MLLMs on P2PEVAL. Specifically, we compare these different model series: GPT (Achiam et al., 2023), Claude (Anthropic, 2024), Doubao (Seed et al., 2025), Qwen (Bai et al., 2025), InternVL (Chen et al., 2024), Gemma (Team et al., 2025), Deepseek (Guo et al., 2025; Liu et al., 2024). Additionally, we include in our evaluation poster images generated by Tencent's AI application, YuanBao (https://yuanbao.tencent.com/), which directly produces academic posters in image format and Chinese. We also fine-tune our model `Qwen3-P2P`, `Qwen2-VL-P2P` and `InternVL3-P2P` using P2PINSTRUCT. The detail of training is shown in Appendix G. And input processing is in Appendix I.

### 4.2 EVALUATION METRICS

Beyond the human-validated Universal Poster Evaluation (max 50) and Fine-Grained Poster Evaluation (max 100) using `GPT-4o` in P2PEVAL, we supplement analysis with objective metrics: (1) ROUGE (Lin, 2004), which measures n-gram overlap between generated and reference poster content, thus capturing lexical similarity. (2) BERTScore (Zhang et al., 2019), which leverages contextual embeddings to assess semantic similarity. During evaluation, all image links are removed from the text to ensure fair comparison of purely textual content. And the "Judge" metric reports how frequently VLLM-based automated evaluators prefer P2P's posters over original author-created versions. We also have aesthetic quality evaluation in Appendix L.

### 4.3 RESULTS AND ANALYSIS

**Main Results.** Table 1 summarizes the experimental outcomes of various models evaluated using P2PEVAL. Our analysis reveals several key findings: **(1) Closed- vs. Open-source Models:** Closed-source models, notably `Claude-3.7-Sonnet`, achieve superior performance in qualitative assessments like the Universal and Fine-Grained evaluation. And leading open-source models such as `Deepseek-R1`(using `Claude` as the provider of Figure Describer), demonstrate strong competitiveness. **(2) Impact of Reasoning Capabilities:** Models employing reasoning or thinking modes such as `Claude-3.7-Sonnet` and `Qwen3` consistently show enhanced performance, especially in the Fine-Grained evaluation. This suggests that advanced reasoning aids in generating outputs that are more aligned with human preferences and detailed content requirements. **(3) Efficacy of P2PINSTRUCT:** Fine-tuning models on P2PINSTRUCT dataset yields substantial and statistically significant improvements (see Appendix H). The `Qwen3-P2P-8B` achieves the highest ROUGE scores across all evaluated models, significantly outperforming its base version and even leading closed-source models in these lexical metrics. It also demonstrates considerable gains in FineGrain and Universal scores over `Qwen3`. The larger gains on lexical metrics like ROUGE reflect the dataset's focus on teaching foundational text generation subtasks, while the consistent improvements on our flagship metrics confirm that these skills translate to higher-level fidelity and quality. These results underscore the value of P2PINSTRUCT. **(4) Supplemental Observations:** A divergence among evaluation criteria is also evident—excellence in lexical overlap (ROUGE) does not uniformly correlate with detailed fidelity (FineGrain), emphasizing the comprehensive nature of P2PEVAL. The strong performance of text-only models utilizing `Claude` for figure description points to the effectiveness of modular, hybrid approaches in this complex generation task.

Table 4: Ablation study results by `Claude-3.7-Sonnet`.

| Mutli Agent | Figure Describer | Reflection | FineGrain | Universal |
|:---:|:---:|:---:|:---:|:---:|
| ✓ | ✓ | ✓ | **65.3962** | **37.2474** |
| ✓ | ✓ | | 64.4556 | 34.2229 |
| ✓ | | ✓ | 63.7388 | 35.1107 |
| ✓ | | | 63.5806 | 33.1458 |
| | | | 60.7233 | 34.2554 |

**Analysis of Human Preference Evaluation.** To complement our P2PEVAL, we conduct pairwise human preference evaluations, the results of which are presented in Table 2. Participants compare posters generated by P2P using `Claude-3.7-Sonnet`, Tencent's YuanBao, and the original author-created posters. The "Preferred or Tied (%)" and "Strictly Preferred (%)" quantify the proportion of instances where method A is judged superior or equivalent to, and strictly superior to, method B, respectively. The results demonstrate a clear preference for P2P-generated posters over those from YuanBao. Notably, P2P also shows competitive performance against original posters, suggesting its capability to produce posters of superior quality in a significant number of cases.

**Analysis of Output Format.** Our investigation of different output formats reveals HTML as the optimal medium for academic posters using `Claude-3.7-Sonnet`. As documented in Table 3, HTML-based poster outputs consistently outperform SVG and LaTeX alternatives across both fine-grained and universal metrics. The inherent flexibility of HTML and CSS for layout structuring and content decoupling, coupled with the robust rendering capabilities of modern browsers, contributes to this performance. Furthermore, our experiments suggest that current LLMs exhibit greater proficiency in HTML code generation compared to equivalent SVG or LaTeX implementations, resulting in fewer rendering errors or structural inconsistencies in the final poster artifacts.

**Ablation Study.** The results of the ablation study of P2P in Table 4 demonstrate that the full system consistently outperforms reduced configurations. When reflection mechanisms (implemented through checker modules) are removed, we observe a moderate decline in universal metrics, suggesting these iterative feedback loops enhance overall poster quality and aesthetic coherence. Similarly, ablating the Figure Describer component, which transforms visual elements into textual descriptions, results in performance degradation. This indicates that directly feeding raw images to MLLMs for content integration can be less effective than providing them with semantically rich textual summaries. These descriptions appear to reduce the interpretative burden on the MLLMs and facilitate a more accurate contextualization of visual information within the poster. Removing all specialized components (resulting in a direct paper-to-poster pipeline without intermediate processing) leads to the greatest performance drop in fine-grained evaluation. This confirms our hypothesis that poster generation benefits significantly from modularized, specialized processing that mimics the distinct cognitive steps humans undertake when creating posters from research papers.

**Analysis of Layout without Reflection.** A comparative analysis of poster layouts generated by Claude and GPT models, summarized in Table 5, reveals distinct structural tendencies inherent in content segmentation and spatial organization for each model when operating without reflection mechanisms. Claude-generated posters typically exhibit a more fragmented structure, utilizing a greater number of columns. These layouts also demonstrate a tendency towards imbalanced spatial distribution, with taller content often concentrated towards the right and greater variability in column heights. This often results in a higher proportion of blank space, suggesting less efficient spatial utilization. In contrast, GPT-generated posters generally present more uniform and compact layouts. These findings suggest challenges in achieving consistent content allocation across the poster layout, a critical aspect for visual appeal and readability in academic posters.

## 5 LIMITATIONS AND FUTURE WORK

While P2P demonstrates significant advances, we acknowledge several limitations that offer avenues for future work.

Table 5: Comparison of layout statistics in posters generated by `Claude-3.7-Sonnet` and `GPT-4o-2024-11-20`.

| Layout Statistic | Claude | GPT | Layout Statistic | Claude | GPT |
|---|---|---|---|---|---|
| **General** | | | **Tallest Column** | | |
| Total columns | 376 | 293 | Height (px) | 7272.22 | 5794.42 |
| **Balance** | | | Text length (char) | 2057.37 | 1554.44 |
| Relative position[1] | 0.55 | 0.51 | Number of images | 2.93 | 2.30 |
| Height coefficient of variation[2] | 0.21 | 0.18 | **Shortest Column** | | |
| Height ratio (max/min)[3] | 1.73 | 1.61 | Height (px) | 4379.82 | 3979.53 |
| Blank space proportion[4] | 19.16% | 14.92% | Text length (char) | 1392.26 | 1296.84 |
| | | | Number of images | 1.74 | 1.59 |

[1] Index of relative column positions within posters; values closer to 0.5 indicate more centered, balanced layouts.
[2] Measure of height consistency across columns; lower values indicate more uniform column heights.
[3] Ratio between tallest and shortest columns; values closer to 1 indicate more even column heights.
[4] Percentage of total poster area occupied by blank space.

- **Output Format:** Our framework currently optimizes for HTML rendering due to its flexibility and the strong capabilities of modern LLMs in generating HTML. As shown in our ablation study (Table 3), generating other formats like LaTeX incurs a performance cost. This presents a practical constraint for users in academic environments where LaTeX or PowerPoint are prevalent. Future work could explore improved code generation for these formats or robust HTML-to-PDF/PPT converters.

- **Computational Cost:** The multi-agent framework with its iterative reflection loops is computationally more intensive than a single-pass generation pipeline. While we demonstrate that the cost is affordable (Appendix K), the latency may be a consideration for real-time applications. The number of reflection iterations serves as a tunable parameter to balance quality and cost.

- **Reasoning Boundaries:** As discussed in Appendix B.2, the checker-reflection mechanism is highly effective at correcting structural and syntactic errors. However, its ability to resolve deep semantic or complex compositional reasoning errors is ultimately bounded by the capabilities of the underlying LLM. For instance, correctly arranging highly intricate, domain-specific multi-panel figures remains a challenge if the base model lacks the necessary visual-spatial reasoning. Our escalation strategy mitigates this but does not eliminate this fundamental boundary.

## 6 CONCLUSION

We introduced P2P, a multi-agent framework that effectively transforms research papers into visually coherent and informationally faithful posters. By modularizing the task and integrating checker-reflection loops, our system demonstrates strong performance, producing outputs that often rival human-created examples. We supported this work by creating P2PINSTRUCT, a large-scale instruction dataset, and P2PEVAL, a comprehensive evaluation benchmark. Crucially, this work argues that evaluating complex, creative artifacts like academic posters requires a dual approach that decouples **objective, verifiable fidelity** from **subjective, holistic quality**. Our P2PEVAL benchmark embodies this principle, combining fine-grained checklists for content preservation with a universal scoring model trained to emulate human aesthetic judgment. This dual-perspective evaluation, alongside the P2P framework and P2PINSTRUCT dataset, not only provides a robust solution for automated poster generation but also offers a principled methodological blueprint for future research in automated scientific communication and other complex creative AI domains. This foundation promises to enhance research accessibility and dissemination efficiency for the entire academic community.

## ETHICS STATEMENT

Our work adheres to the principles outlined in the ICLR Code of Ethics. The primary goal of this research is to contribute to societal and human well-being by developing tools that streamline

and accelerate scholarly communication, making research more accessible and saving researchers valuable time.

In line with the principle of upholding high standards of scientific excellence and avoiding harm, we acknowledge the potential risk of automated systems misrepresenting or "hallucinating" scientific content. To mitigate this, our core contribution includes the P2PEVAL benchmark, which is specifically designed to evaluate the factual fidelity of generated posters. Our dual-evaluation framework, combining Fine-Grained checklist-based verification for objective accuracy and a Universal score for holistic quality, represents a direct effort to ensure the trustworthiness and reliability of the output.

To respect the work required to produce new ideas and artefacts, all papers and posters used to construct our P2PEVAL benchmark were sourced from publicly available repositories (ACL, Sci-PostLayout) under permissive licenses (CC-BY, CC4.0), and we give full credit to the original authors.

In developing our benchmark and datasets, we engaged human annotators. We are committed to fairness and respect for all individuals involved. As detailed in Appendix C, our annotation team consisted of qualified domain experts who were compensated at a rate exceeding the local statutory minimum wage. We established a rigorous annotation protocol, including training and verification steps, to ensure data quality and consistency, while also protecting the privacy and well-being of the participants.

We acknowledge the environmental cost associated with training large models and have reported our resource consumption in Appendix J. We believe the potential benefits of our work in enhancing the efficiency of scientific dissemination for the broader research community justify this computational cost. Finally, we discuss the limitations of our system in Section Q to provide a transparent account of its current capabilities.

## REPRODUCIBILITY STATEMENT

To ensure the reproducibility of our work, we have made comprehensive efforts to provide all necessary components. All code for the P2P multi-agent framework, the P2PEVAL evaluation pipeline, and the fine-tuning process will be made publicly available in the anonymous GitHub repository linked in the paper.

The datasets created for this research will also be released. This includes P2PINSTRUCT, our large-scale instruction-tuning dataset of over 30,000 examples, and P2PEVAL, which comprises 121 paper-poster pairs and 1738 human-annotated checklist items in a structured YAML format.

The paper provides detailed descriptions of our methodology. The architecture of the P2P framework is detailed in Section 2 and illustrated in Figure 1. The construction of the P2PEVAL benchmark and the full evaluation protocol are described in Section 3. For implementation-specific details, Appendix G documents the fine-tuning hyperparameters and setup using the LLaMA-Factory framework. Appendix I describes the input processing pipeline for both text and figures. Furthermore, we include the exact prompts used for each agent and evaluation step in the Appendix to allow for full replication of our generation and evaluation logic.

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

# Appendix

## CONTENTS

# A    RELATED WORK

**Poster Generation.**    Academic poster generation involves creating a poster that summarizes the key information from an academic paper. Paramita et al. Paramita & Khodra (2016) develop a model that extracts essential sentences into templates to generate text-based posters. Qiang et al. Qiang et al. (2019) propose a more comprehensive method, decomposing poster generation into three subtasks: content extraction Mihalcea & Tarau (2004); Xu & Wan (2021); Cheng et al. (2024b;a), panel attribute inference Zhong et al. (2019); Li et al. (2020); Huang et al. (2022), and panel layout generation Lin et al. (2024); Zhang et al. (2023). Postdoc Jaisankar et al. (2024) utilizes MLLMs to generate template-based posters but cannot produce flexible layouts with more dynamic integration of figures and text. Additionally, existing academic poster datasets Yao et al.; Xu & Wan (2021); Qiang et al. (2016); Wang et al. (2024a); Saxena et al. (2025) lack fine-grained evaluation metrics necessary for comprehensive quality assessment.

**HTML Code Generation and Multi-Agent.**    Recent research in automated front-end development focuses on generating HTML from diverse inputs such as screenshots, prototypes and natural language. This has spurred the creation of benchmarks like Design2Code (Si et al., 2024; Yang et al., 2025), Websight (Laurençon et al., 2024), WebCode2M (Gui et al., 2025a), and Web2Code (Yun et al., 2024). Code generation methodologies vary, including direct translation, structured approaches such as DCGen's (Wan et al., 2024b) divide-and-conquer strategy and UICopilot's (Gui et al., 2025b) hierarchical generation. Applications target mobile UIs (Xiao et al., 2024b; Zhou et al., 2024), multi-page websites (Wan et al., 2024a), and web design (Xiao et al., 2024a; Li et al., 2024; Zhang et al., 2024), with model fine-tuning (Liang et al., 2024) enhancing performance. Multi-agent systems are increasingly adopted for complex tasksHan et al. (2024); Liu et al. (2025); for instance, agentic workflows can convert designs to code (Ding et al., 2025; Islam et al., 2024), and some systems employ distinct agents for sub-tasks with iterative human feedback(Wang et al., 2024b).

**LLM as a Judge.**    The use of LLMs as evaluators, termed "LLM-as-a-Judge," is well-studied and has demonstrated high consistency with human judgment, with early work focusing on LLMs evaluating other LLMs, as seen in JudgeLM Zhu et al. (2023); Yang et al. (2024a). Subsequent research introduced systems like AUTO-J Li et al. (2023a), leveraging pairwise and single-response evaluations to achieve strong agreement with human assessments Bai et al. (2023); Li et al. (2023b); Yang et al. (2024b); Li et al. (2023c); Sun et al.; Yang et al. (2021); Sun et al. (2025). With the rise of MLLMs, their potential as evaluators in multimodal tasks is being explored, as traditional metrics often fail to capture the nuances of complex multimodal outputs Antol et al. (2015); Liu et al. (2023a;d;c;b). To enhance LLM evaluation capabilities, techniques such as Chain-of-Thought Wei et al. (2021); Chu et al.; Chai et al. (2025); Sun et al. (2024) and Training-free instruction following Brown et al. (2020); Wei et al. (2021) have been proposed, addressing the need for more robust evaluators in both unimodal and multimodal contexts.

# B    MORE DETAILS OF P2P AND IMPLEMENTATION

## B.1    AGENT IMPLEMENTATION AND MODEL AGNOSTICISM

As stated in the main paper, P2P is a model-agnostic orchestration framework. For any given experiment in Table 1, all core generative components (the Figure Describer, Section Generator, Content Generator, and HTML Generator) are powered by the same underlying LLM being evaluated. This experimental design ensures a fair, end-to-end benchmark of each model's capabilities on the complex paper-to-poster task. The only fixed components (shown in Appendix I) are programmatic utilities like PyMuPDF for text extraction and DocLayout-YOLO for initial figure detection.

## B.2    CHECKER-REFLECTION MECHANISMS IN DETAIL

The checker-reflection mechanism is central to P2P's quality assurance. As shown in Fig 5, here we detail each checker's implementation, failure triggers, and the nature of the reflection loop.

**Figure Checker.**    This module is rule-based and programmatic for efficiency. Its reflection loop is triggered if the count of detected figures mismatches the count of detected captions (paired using

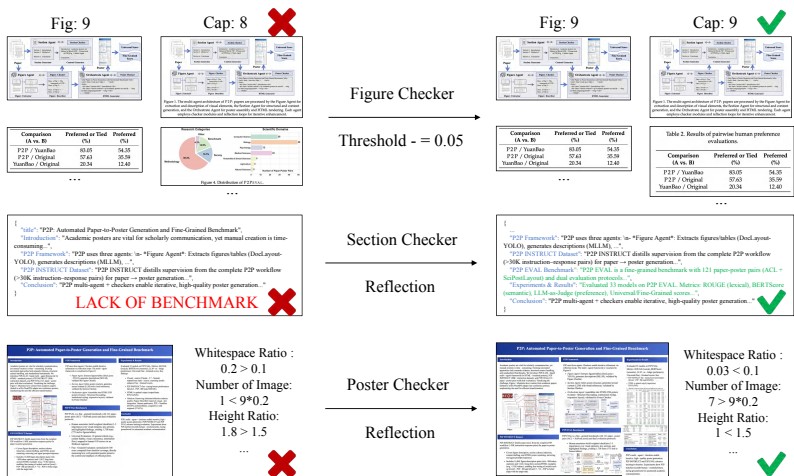

Figure 5: Illustration of the Checker-Reflection Mechanisms.

Manhattan distance). The initial confidence threshold for the DocLayout-YOLO detector is set to a high 0.85 to minimize false positives. If a mismatch occurs, the threshold is iteratively decreased by 0.05. This allows the Figure Extractor to capture elements that may have been initially assigned a lower confidence score (e.g., tables that visually resemble plain text), ensuring all significant visuals are reliably captured.

**Section Checker.**    This module uses an LLM-as-a-Judge to assess semantic qualities. After content generation, the checker prompts the LLM to evaluate the text against the source paper on coherence, completeness, faithfulness, and correct referencing. The LLM returns a binary decision ("OK" or "Problem"). If a problem is detected, it also provides actionable feedback for revision (e.g., "The conclusion section omits the key limitation mentioned in the paper."). The Content Generator then re-runs with this new feedback, ensuring iterative refinement.

**Poster Checker.**    This module is also rule-based. It programmatically parses the generated HTML to check for structural integrity and layout aesthetics. For example, a reflection loop is triggered if the **proportion of blank space exceeds 10%** or if column heights are severely imbalanced. The checker then feeds these failed metrics (e.g., '"whitespace_ratio": 0.15') and previous HTML code back to the HTML Generator, instructing it to regenerate the layout. On average, this loop runs for 2.32 iterations for GPT-4.1 and 3.47 for Qwen-2.5-VL-7B.

**Failure Boundaries and Escalation.**    The reflection paradigm is highly effective at correcting structural and syntactic errors. However, it can struggle with deep semantic or complex compositional reasoning errors. For instance, if a paper contains numerous, intricate multi-panel figures that require a highly specific arrangement to be coherent, the LLM might struggle to generate a logical HTML structure. The Poster Checker may detect the resulting layout imbalance (e.g., high whitespace), but if the root cause is the LLM's inability to reason about the complex spatial relationship between sub-figures, reflection alone may not solve it. The LLM, lacking a better strategy, might simply reshuffle the same flawed components. To mitigate this, we employ an **escalation strategy**: if a checker fails after 5 retries, the reflection loop can escalate to an earlier agent, prompting a more fundamental revision. For example, a persistent layout failure in the Orchestrate Agent can trigger the Section Agent to regenerate its content entirely, attempting to provide a simpler or more structured input that the Orchestrate Agent can handle. This provides a multi-level recovery mechanism, though it is ultimately still bounded by the LLM's core reasoning ability.

## C    ANNOTATOR TEAM

Our team comprised 12 annotators, all holding university degrees, with six possessing research-based master's degrees and six holding doctorates. This composition guaranteed that for any given paper, at

least three annotators possessed relevant domain expertise. Before the formal rating process, each annotator completed a one-hour training session.

We have identified the following potential risks and their corresponding mitigation measures:

1. **Privacy and Confidentiality Risks:** The sharing of personal information by participants may lead to privacy breaches. To mitigate this, all data will be anonymized and stored securely, with access restricted to authorized research personnel only.

2. **Psychological Risks:** Participants may experience discomfort or stress during task execution. To address this, we provide detailed task instructions and debriefing sessions to ensure participants feel supported throughout the process. Additionally, participants have the right to withdraw from the study at any stage without penalty.

3. **Physical Risks:** Although our research procedures do not involve significant physical risks, we will closely monitor participants for any signs of distress and provide necessary support promptly.

Furthermore, regarding data annotation, we have paid annotators a wage higher than the statutory minimum wage in the country of the data annotators, as a sign of respect for their labor.

## D  MORE DETAILS OF P2PINSTRUCT

**Data Source and Generation.** P2PINSTRUCT was constructed using the training split of the SciPostLayout dataset, which is separate from our test set. We chose this source for its permissive CC-BY license and topic diversity. And we avoided using papers from sources like ICLR after consulting with legal professionals, as their copyrights would require author-by-author permissions. To mitigate the risk of "imprinting bias" from a purely synthetic loop, we used a **"teacher-forcing-like" approach**. At each stage of data generation, the model was guided by the corresponding part of the original, human-created poster. For example, the HTML Generator received the generated Markdown but was also shown the ground-truth poster's image as a visual layout reference. This process anchors the synthetic data to human design principles.

**Quality Validation.** The "high-quality" claim is supported by three pillars: (1) **Built-in Quality Control** from the checker-reflection loops during generation; (2) **Manual Verification** of a random sample of 20 pairs from each stage; and (3) **Empirical Validation**, where models fine-tuned on P2PINSTRUCT consistently and significantly outperform their base versions (Table 1), providing strong evidence of the dataset's effectiveness.

## E  MORE DETAILS OF P2PEVAL

### E.1  CHECKLIST FORMAT

All checklist annotations, including unique paper identification, detailed evaluation criteria, reference figures (when applicable), and established maximum scores, are documented in YAML format.

Each item is a distinct, verifiable component with a description, figure, and max score. For clarity, here is an example that is similar to the following simplified version:

```yaml
name: paper_id
checklist:
 - description: Does the introduction section highlight the limitations
   of current methods in code generation tasks and motivate the use of
   xxxxx?
  max_score: 4
 - description: Does the introduction section provide xxxxx examples for
   visual demonstration?
  figure: 0
  max_score: 5
```
Listing 1: A YAML checklist for paper evaluation.

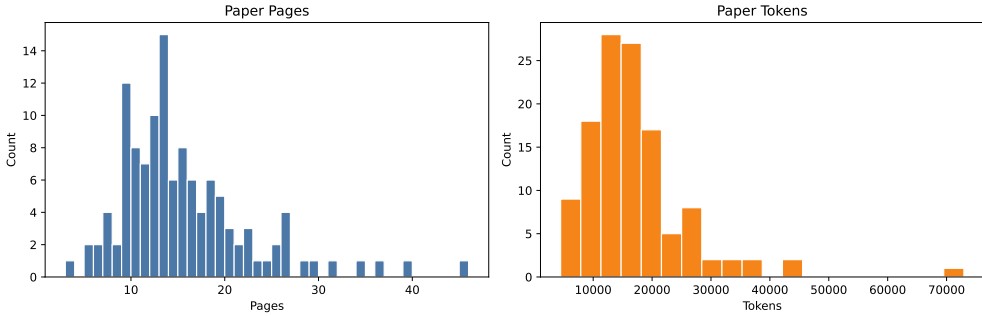

Figure 6: Distributions of Paper

Among them, xxxxx refers to what was proposed in paper_id, which has been anonymized for double-blind review; figure: 0 indicates there is a reference image (the image is taken from the original paper); max_score represents the weight assigned by human annotators. More examples are provided in the anonymous GitHub repository linked to supplementary materials.

## E.2 BENCHMARK STATISTICS

P2PEVAL is composed of 121 complete paper-poster pairs. Each sample in the benchmark is comprehensive, including the original research paper ('.pdf'), the author-created poster ('.pdf' and '.png'), and our detailed, human-annotated 'checklist.yaml' file. The statistics presented in Table 6 highlight the complexity and diversity of the data, underscoring the challenges of the paper-to-poster task.

Table 6: Key statistics of the P2PEVAL benchmark dataset.

| Metric | Value |
|---|---|
| Total Paper-Poster Pairs | 121 |
| Total Checklist Items | 1,738 |
| Total Visual Elements in Checklists | 775 |
| **Average per Sample** | |
| Paper Length (Pages) | 15.13 |
| Paper Content (Tokens) | 17,104 |
| Poster Content (Tokens) | 2,050 |
| Checklist Content (Tokens) | 548 |
| Images per Sample | 7.25 |

The source documents are substantial academic works, with an average length of 15.1 pages and approximately 17,100 tokens per paper. The corresponding posters are significantly more concise, averaging around 2,050 tokens. This reflects a textual summarization ratio of approximately 8.3-to-1, demonstrating the extensive content distillation required for poster generation.

The multimodal nature of the task is evident from the data. On average, each sample contains 7.25 visual elements (figures and tables), with some papers including as many as 28. The richness of our annotations is highlighted by the 1,738 unique checklist items across the dataset, averaging 548 tokens of descriptive metadata per sample. This fine-grained annotation provides a robust foundation for the detailed evaluation of model-generated posters. The distribution histograms 6 show that the

majority of papers range from 8 to 17 pages, and most contain between 5 and 10 images, representing a typical cross-section of academic publications.

## E.3 ANALYSIS OF PERFORMANCE BREAKDOWN BY TOPIC

Table 7: Performance breakdown by research category.

| Metric | Methodology | Survey | Benchmark | Other |
|---|---|---|---|---|
| Fine-Grained Score | 64.24 | 65.88 | **68.78** | 65.92 |
| Universal Score | 42.21 | 40.95 | **42.46** | 40.63 |

Table 8: Performance breakdown by scientific field.

| Metric | Comp. Sci. | Biology | Psychology | Medical Sci. | Hum. & Soc. Sci. |
|---|---|---|---|---|---|
| Fine-Grained Score | 66.59 | 64.17 | **73.42** | 66.51 | 62.84 |
| Universal Score | **42.80** | 42.21 | 41.45 | 39.85 | 39.64 |

We analyze the performance of our best-performing model (Claude-3.7-Sonnet) across different research categories and scientific fields from our P2PEVAL.

- **Performance by Research Category**: The tab 7 shows that our P2P performs exceptionally well on Benchmark papers, which typically have a highly regular structure (e.g., Task Definition, Dataset, Results). This structure is more easily parsed and summarized by LLMs. Methodology and Survey papers, which can be more conceptually dense, also yield strong results, demonstrating the model's versatility.

- **Performance by Scientific Field**: The tab 8 shows that P2P achieves the highest Fine-Grained score in Psychology, likely because papers in this field often use standardized charts (bar graphs, scatter plots) that are well-understood by MLLMs. Computer Science papers achieve the highest Universal score, potentially due to the abundance of CS papers in training data. The slightly lower scores in Humanities & Social Sciences may be due to the reliance on denser, narrative text over structured figures, highlighting a potential area for future improvement.

## F MORE DETAILS OF UNIVERSAL POSTER EVALUATION

### F.1 DETAILS OF THE HUMAN RATING PROTOCOL

As stated in the main paper, our Universal Poster Evaluation score is not directly generated by an LLM. Instead, it is predicted by an XGBoost model trained to replicate human judgment. This was a deliberate design choice, as we found that while LLMs can reliably score discrete, objective criteria, they struggle to capture the complex, non-linear weightings that humans apply when forming a single holistic impression of a poster's quality. The following protocol was used to collect the human scores that serve as ground truth labels for training this XGBoost model.

**Annotator Qualifications and Training.** Details of our annotation team are shown in Appendix C. During the training session, we reviewed the 10 universal criteria (detailed in Section F.2) and provided scoring guidelines using concrete examples. For instance, award-winning NeurIPS, ICML and ICLR posters were presented as examples deserving a full score (50/50), while posters with significant clarity issues, poor visual design, or logical gaps were used to demonstrate how deductions should be made.

**Rating Procedure and Scale.** For Universal Poster Evaluation, annotators produced a single holistic score in [0, 50], referencing U1–U10. We provided explicit anchor points to reduce scale ambiguity:

- **45–50: Outstanding.** Meets or exceeds expectations on most criteria; highly professional.
- **35–44: Strong.** Minor issues on a few criteria; clearly presentation-ready.
- **25–34: Adequate.** Several weaknesses, but overall understandable and usable.
- **15–24: Weak.** Multiple violations (e.g., poor fidelity, cluttered layout, weak flow).
- **0–14: Poor.** Severe issues (e.g., illegible, misleading, not self-contained).

Each poster was rated independently by three annotators. Similar to the annotation of the checklist, any posters with substantial initial scoring disagreements were excluded from the final training set for the XGBoost model to maintain high data quality. We utilize both powerful models like GPT-4o and lighter models such as Qwen-VL-2.5-32B, ensuring the trained annotators are exposed to diverse samples to enhance generalizability. We collected a total of 1,701 human ratings across the dataset. To avoid any leakage when training the scoring model, these ratings were collected on outputs produced by an ablated system variant without the checker-reflection mechanism.

**Reliability and Agreement.** The rigor of our protocol is validated by strong inter-annotator agreement statistics. We achieved a Krippendorff's Alpha of **0.95**, a Spearman correlation of **0.96** and Cohen's Kappa of 0.71, indicating exceptional reliability and consistency among our raters' judgments. These high-quality human scores formed the ground-truth labels ($y$) used to train our final XGBoost scoring model. We also observed low divergence between predicted and human score distributions (KL divergence: 0.1093; JS divergence: 0.0239), supporting the reliability of our scoring pipeline.

### F.2 DETAILS OF EVALUATION CRITERIA

The 10 universal criteria form the input feature vector for our XGBoost scoring model. For each generated poster, an LLM-as-a-Judge (GPT-4o) is prompted to score the poster on each of the following 10 dimensions on a discrete scale from 0 to 5. Below are the detailed descriptions for each criterion as used in the evaluation prompt.

$U_1$**: Authorship and Title Accuracy** *Description:* Does the poster clearly and accurately display the complete paper title and the full names of all authors without any spelling errors, omissions, or formatting mistakes? *Scoring Guideline:* A score of 5 requires a perfect match. Deductions are made for typos, missing authors, or an incomplete title.

$U_2$**: Image Uniqueness and Quality** *Description:* Are all images in the poster unique (i.e., no unintended duplications)? Is the visual quality of each image (resolution, clarity) sufficient for a professional presentation? *Scoring Guideline:* A score of 5 requires all images to be distinct and high-resolution. Points are deducted for blurry, pixelated, or duplicated figures.

$U_3$**: Balanced White Space** *Description:* Is the negative or "white" space distributed effectively across the poster? Does the layout avoid areas that look visually overcrowded or, conversely, excessively empty? *Scoring Guideline:* A high score indicates a layout that feels balanced and guides the eye naturally.

$U_4$**: Contextual Relevance** *Description:* Do the visual elements (figures, tables) align logically with the adjacent text? Is their placement and thematic connection clear, enhancing the reader's comprehension? *Scoring Guideline:* A score of 5 means every figure is placed next to the text that describes or references it.

$U_5$**: Optimal Visual-to-Text Ratio** *Description:* Does the proportion of the poster's area covered by images effectively serve the research content? Is there a good balance between visual evidence and textual explanation? *Scoring Guideline:* The optimal ratio is context-dependent, but a high score reflects a poster that uses visuals impactfully without overwhelming the text, or vice-versa.

$U_6$**: Dimension Appropriateness** *Description:* Are the overall dimensions (width and height) of the poster suitable for its content and a typical presentation environment? Does it avoid extreme aspect ratios that would make it difficult to read or display? *Scoring Guideline:* A score of 5 indicates standard poster dimensions (e.g., portrait A0, landscape 48"x36"). Extreme, banner-like shapes receive lower scores.

**U$_7$: Visual Consistency** *Description:* Do the design elements—such as color schemes, typography (fonts, sizes), and section heading styles—maintain a cohesive and consistent identity throughout the poster? *Scoring Guideline:* A high score is given for a poster with a unified visual theme. Using a chaotic mix of fonts or colors results in a low score.

**U$_8$: Content Fidelity** *Description:* Are the data representations, mathematical formulas, key terminology, and scientific findings presented on the poster identical to those in the original research paper? *Scoring Guideline:* A score of 5 requires zero "hallucinated" or misrepresented facts. Any deviation from the source paper's content results in a score deduction.

**U$_9$: Information Flow Logic** *Description:* Is the content organized in a logical and intuitive sequence (e.g., Introduction → Methods → Results → Conclusion)? Can a viewer easily follow the research narrative from start to finish? *Scoring Guideline:* High scores are for posters with a clear, linear, or grid-based flow. Disorganized or confusing layouts receive low scores.

**U$_{10}$: Self-Contained Explanation** *Description:* Can the poster be fully understood on its own, without requiring a verbal explanation from a presenter? Does it provide enough context and detail for a knowledgeable reader to grasp the core concepts and contributions? *Scoring Guideline:* A score of 5 indicates a poster that is fully self-sufficient. A poster that is just a collection of figures with cryptic captions would score poorly.

## F.3 Other Methods Performance

Our choice of a hybrid MLLM-Featurizer and XGBoost model for the Universal Score was a principled design decision to create a more robust and human-aligned evaluation. Directly asking an LLM for a single holistic score is often noisy and inconsistent. Our approach combines the strengths of different models:

- **Stable Feature Extraction (LLM):** We use the LLM for what it does best: scoring 10 discrete, well-defined criteria (U1-U10).
- **Learning Human Preferences (XGBoost):** We use XGBoost to learn the complex, non-linear function of how humans weigh these criteria, training it on 1,701 human preference scores.

To empirically validate this, we conducted an ablation study comparing our method against simpler baselines. As shown in Table 9, our hybrid approach is significantly more aligned with human judgment than end-to-end LLM scoring based on Qwen-2.5-VL-32B.

Table 9: Ablation study on the Universal Score methodology, validating the choice of the MLLM-Featurizer and XGBoost approach. Metrics are correlations ($R^2$) and distributional similarity (KL/JS Divergence, lower is better) against human scores.

| Method | $R^2$ | KL-Divergence | JS-Divergence |
|---|---|---|---|
| Direct LLM Scoring (End-to-End) | 0.27 | 1.60 | 0.62 |
| LLM as Regressor (Replaces XGBoost) | 0.51 | 0.34 | 0.22 |
| Fine-Tuned LLM as Aggregator | 0.70 | **0.06** | 0.14 |
| **Our MLLM-Featurizer + XGBoost** | **0.92** | 0.11 | **0.02** |

Additionally, we experiment with other methods, including Ordinary Least Squares($R^2 = 0.66$), Random Forest($R^2 = 0.83$), and various regularization techniques($R^2 = 0.89$); however, these approaches yield suboptimal performance compared to XGBoost.

## G Training Details

We utilized the unified and efficient LLaMAFactory framework (Zheng et al., 2024) for all fine-tuning experiments. We use our P2PINSTRUCT dataset with a learning rate of $5 \times 10^{-5}$ for 3 epochs and employing AdamW (Loshchilov & Hutter, 2017). To ensure optimal instruction-following, we use

the native chat template for each respective model (e.g., Qwen3's template) . Training is conducted with BF16 mixed-precision to accelerate the process. No sequence packing was used, and we set a maximum sequence length of 8000, truncating longer examples.

## H    STATISTICAL SIGNIFICANCE OF RESULTS

We performed statistical significance testing for our key experimental comparisons. Following modern best practices (Wasserstein & Lazar, 2016; Jeppesen et al., 2021; Brunoni et al., 2013), we report **Effect Size (Cohen's d)**, which quantifies the practical magnitude of an observed difference. A large effect size ($|d| > 0.8$) indicates a meaningful, substantial improvement. As shown in Table 10, fine-tuning on P2PINSTRUCT yields a large effect size, and our ablation study confirms that removing key components has a medium-to-large negative effect on quality.

Table 10: Effect size (Cohen's d) for fine-tuning and ablation studies. Positive values indicate improvement over the baseline; negative values indicate degradation.

| Comparison | FineGrain (d) | Universal (d) |
|---|---|---|
| *Effect of Fine-Tuning on* P2PINSTRUCT | | |
| InternVL3-P2P vs. Base | 2.13 | 1.72 |
| Qwen2.5-VL-P2P vs. Base | 4.18 | 3.00 |
| Qwen3-P2P vs. Base | 1.21 | 0.60 |
| *Effect of Ablating Components (vs. Full* P2P) | | |
| w/o Reflection | -0.15 | -0.61 |
| w/o Figure Describer | -0.26 | -0.43 |
| Multi-Agent Only | -0.28 | -0.82 |
| End-to-End Only | -0.72 | -0.60 |

## I    INPUT PROCESSING

All models are configured with the temperature of 1 and the maximum output token length of 8000 to ensure fair comparison while maintaining generation diversity. To isolate the benefit of our architecture from backbone capacity, we also include text-only backbones that consume pre-extracted figure descriptions(using `Claude-3.7-Sonnet` as the provider of Figure Describer).

And all models receive the same inputs:

- PDF text extracted with PyMuPDFLoader(https://github.com/pymupdf/PyMuPDF) for Section Agent.
- Figures/tables detected with DocLayout-YOLO (Zhao et al., 2024) for Figure Agent.
- For text-only LLMs, which cannot process images, figure descriptions are generated using `Claude-3.7-Sonnet` as the provider of Figure Describer.

## J    RESOURCE CONSUMPTION

The training phase consumes GPU compute equivalent to 80 A100-GPU hours, whereas evaluating multiple models via the APIs offered by our provider, OpenRouter and Volcano Engine, incurs a cost of more than $500.

## K    COMPUTATIONAL COST AND LATENCY

To help users assess practical deployability, we provide a detailed cost and latency analysis for generating a poster for an average paper (based on Table 6: 17,104 text tokens and 7.25 images). We report metrics for GPT-4.1 (representing powerful closed-source models) and Qwen-2.5-VL-7B (representing efficient open-source models). The average runtimes, using multi-threading and parallel

optimizations, are 209.32 seconds for GPT-4.1 and 55.61 seconds for Qwen-2.5-VL-7B. Below is an approximate breakdown of token consumption and cost, accounting for the retry factors measured during our experiments. The cost of the Figure Extractor (YOLO detection) is excluded as it is a fixed, local computation.

**GPT-4.1 Breakdown**

- **Figure Agent:** Avg. input = 391.84 (image tokens) $\times$ 7.25; Avg. output = 206.40 $\times$ 7.25.
- **Section Agent:** Avg. input = $(17,104 \times 2) + 338$ (prompt) $+ (206.40 \times 7.25)$; Avg. output = $2,158.84$. This stage has a retry factor of $1.05\times$.
- **Orchestrate Agent:** Avg. input = $17,104 + 2,158.84 + 1,145$ (prompt); Avg. output = $2,505.84$. This stage has a retry factor of $2.32\times$.
- **Total:** This results in an average total of **88,031 input tokens** and **9,577 output tokens**.

Based on OpenRouter pricing for GPT-4.1 ($2/M input, $8/M output):
Cost = $(88,031/1,000,000 \times \$2) + (9,577/1,000,000 \times \$8) \approx \$\mathbf{0.25}$ per poster.

**Qwen-2.5-VL-7B Breakdown**

- **Figure Agent:** Avg. input = 391.84 (image tokens) $\times$ 7.25; Avg. output = 148.96 $\times$ 7.25.
- **Section Agent:** Avg. input = $(17,104 \times 2) + 338$ (prompt) $+ (148.96 \times 7.25)$; Avg. output = $1,479.50$. This stage has a retry factor of $1.03\times$.
- **Orchestrate Agent:** Avg. input = $17,104 + 1,479.50 + 1,145$ (prompt); Avg. output = $1,637.47$. This stage has a retry factor of $3.47\times$.
- **Total:** This results in an average total of **107,993 input tokens** and **8,286 output tokens**.

Based on OpenRouter pricing for Qwen-2.5-VL-7B ($0.20/M for both input/output):
Cost = $(107,993/1,000,000 \times \$0.20) + (8,286/1,000,000 \times \$0.20) \approx \$\mathbf{0.02}$ per poster.

This analysis demonstrates that P2P offers a tunable cost-performance trade-off and is practically affordable for users, with costs ranging from a couple of cents to a quarter per poster depending on the chosen backbone model.

## L  AESTHETIC QUALITY EVALUATION

To complement our existing metrics, we introduce a quantitative evaluation of aesthetic principles standard in graphic layout analysis (Deng et al., 2017; Datta et al., 2006; Lu et al., 2015; Marchesotti et al., 2011). We analyze the generated posters using a suite of metrics measuring alignment, balance, symmetry, and other design fundamentals. As shown in Table 11, our best-performing model produces layouts with strong balance and color harmony, providing an objective measure of aesthetic quality.

Table 11: Quantitative aesthetic evaluation of generated posters. Results show mean $\pm$ 95% CI. Higher is generally better. Best results are in bold.

| Metric | Higher is better? | Claude-3.7-Sonnet | GPT-4o-mini-2024-07-18 |
|---|---|---|---|
| Alignment | Yes | $0.455 \pm 0.005$ | $\mathbf{0.476 \pm 0.007}$ |
| Balance (Left/Right) | Yes | $\mathbf{0.894 \pm 0.014}$ | $0.744 \pm 0.032$ |
| Balance (Top/Bottom) | Yes | $\mathbf{0.790 \pm 0.022}$ | $0.784 \pm 0.020$ |
| Symmetry | Yes | $\mathbf{0.573 \pm 0.016}$ | $0.447 \pm 0.026$ |
| Whitespace Ratio | Moderate (target-dependent) | $0.719 \pm 0.007$ | $0.684 \pm 0.015$ |
| Rule of Thirds | Yes | $0.666 \pm 0.015$ | $\mathbf{0.668 \pm 0.017}$ |
| Contrast (RMS) | Yes | $\mathbf{0.257 \pm 0.007}$ | $0.247 \pm 0.009$ |
| Color Harmony | Yes | $\mathbf{0.908 \pm 0.010}$ | $0.846 \pm 0.014$ |
| Simplicity (1 - Clutter) | Yes | $\mathbf{0.784 \pm 0.004}$ | $0.779 \pm 0.007$ |

We define each metric used in our evaluation as follows:

**Alignment:** This metric quantifies the proportion of strong lines in the image that are aligned with the horizontal and vertical axes. It is calculated by first computing the image gradients using a Sobel filter. Then, it measures the magnitude-weighted sum of pixels whose gradient direction is within a small tolerance ($\pm10^\circ$) of 0, 90, or 180 degrees. This sum is normalized by the total gradient magnitude across the entire image. A high score indicates a strong presence of structured horizontal and vertical elements, which is a key principle of organized design.

**Balance (Left/Right):** This measures the distribution of visual weight between the left and right halves of the image. Visual weight is approximated by the density of edges detected by a Canny edge detector. The score is calculated as $1 - \frac{|E_L - E_R|}{E_L + E_R}$, where $E_L$ and $E_R$ are the sum of edge pixel intensities in the left and right halves, respectively. A score of 1.0 signifies perfect left-right balance.

**Balance (Top/Bottom):** Similar to Left/Right Balance, this measures the equilibrium between the top and bottom halves of the image based on edge distribution. The score is calculated as $1 - \frac{|E_T - E_B|}{E_T + E_B}$, where $E_T$ and $E_B$ are the sum of edge pixel intensities in the top and bottom halves. A score of 1.0 signifies perfect top-bottom balance.

**Symmetry:** This metric assesses horizontal reflectional symmetry. It is computed by calculating the normalized cross-correlation (cosine similarity) between the grayscale left half of the image and a horizontally flipped version of the right half. The score ranges from 0 to 1, where 1 indicates perfect symmetry between the two halves.

**Whitespace Ratio:** This metric calculates the proportion of the image area that constitutes the background or "whitespace." The background color is estimated by taking the median color of the image's border pixels in the perceptually uniform CIELAB color space. The score is the fraction of total pixels that are perceptually close (within a fixed threshold) to this estimated background color.

**Rule of Thirds:** This evaluates the composition's adherence to the Rule of Thirds. It first identifies the center of mass (centroid) of the image's edge content. It then measures the shortest distance from this centroid to one of the four "power points" (intersections of the rule-of-thirds grid). The score is inversely proportional to this distance, normalized so that a centroid located directly on a power point yields a high score.

**Contrast (RMS):** This measures the global contrast of the image, calculated as the Root Mean Square (RMS) contrast. It is defined as the standard deviation of the pixel intensities in the grayscale version of the image, normalized to a [0, 1] range. A higher value indicates a wider dynamic range and stronger tonal separation.

**Color Harmony:** This metric, derived from the 'hue_peak_score', assesses the simplicity and harmony of the color palette. It computes a histogram of the image's hue channel and sums the normalized frequencies of the two most dominant hues. A high score suggests the palette is concentrated around a small number of primary hues, often leading to a more cohesive and harmonious visual experience.

**Simplicity (1 - Clutter):** This metric quantifies the visual simplicity of the poster by scoring the inverse of clutter. Clutter is defined as a weighted sum of two components: 1) structural complexity, measured by the overall density of edges, and 2) color complexity, measured by the entropy of the image's saturation and value (brightness) channels. The final score is $1 - \text{clutter}$, where a higher value corresponds to a cleaner, less cluttered design.

## M THE FEATURES OF FINE-GRAINED POSTER EVALUATION

The Fine-Grained Poster Evaluation pipeline offers several distinct advantages:

1. **Ground-Truth Alignment**: Each checklist item references specific elements from the official academic posters and corresponding papers, ensuring accurate evaluation aligned with the original author's intent.

2. **Domain-Specific Emphasis**: The pipeline captures domain-specific expectations and conventions, which universal criteria may overlook, reflecting discipline-specific priorities.

3. **Essential Research Component Verification**: Critical content such as key figures, methodology details, and conclusions is explicitly accounted for using detailed scoring mechanisms, ensuring comprehensive evaluation.

4. **Human Preference Integration**: Carefully calibrated by four human annotators, checklist item scores inherently encode domain expertise and human judgment regarding item significance and presentation quality.

## N   THE FEATURES OF HTML FORMAT

We compare the advantages of HTML for SVG and LaTeX:

- **Universal Accessibility and Portability**: HTML posters can be easily viewed on any device with a web browser, requiring no specialized software (unlike LaTeX, which needs compilation, or potentially specific viewers for complex SVGs).

- **Rich Interactivity**: HTML, often combined with CSS and JavaScript, allows for the seamless integration of interactive elements such as hyperlinks (to papers, datasets, author profiles), tooltips, expandable sections, or even embedded multimedia. This level of interactivity is more cumbersome to achieve and less natively supported in LaTeX or static SVG.

- **Flexible and Modern Styling**: CSS provides powerful and flexible control over the visual presentation, enabling modern, responsive, and aesthetically engaging designs that can adapt to various screen sizes. This offers more design freedom than typical LaTeX layouts and better structural organization for complex content than a single SVG.

- **Ease of Web Integration**: As the native language of the web, HTML posters can be effortlessly embedded into websites, shared via links, and are inherently well-suited for online conference platforms and digital dissemination.

## O   EXAMPLES OF POSTER GENERATION

Examples of poster generation are shown in Fig 7, Fig 8 and so on. A wide range of examples features spanning different formats (e.g., landscape, portrait, multi-column, spanning columns) and scientific disciplines, effectively highlighting the P2P's flexibility and versatility.

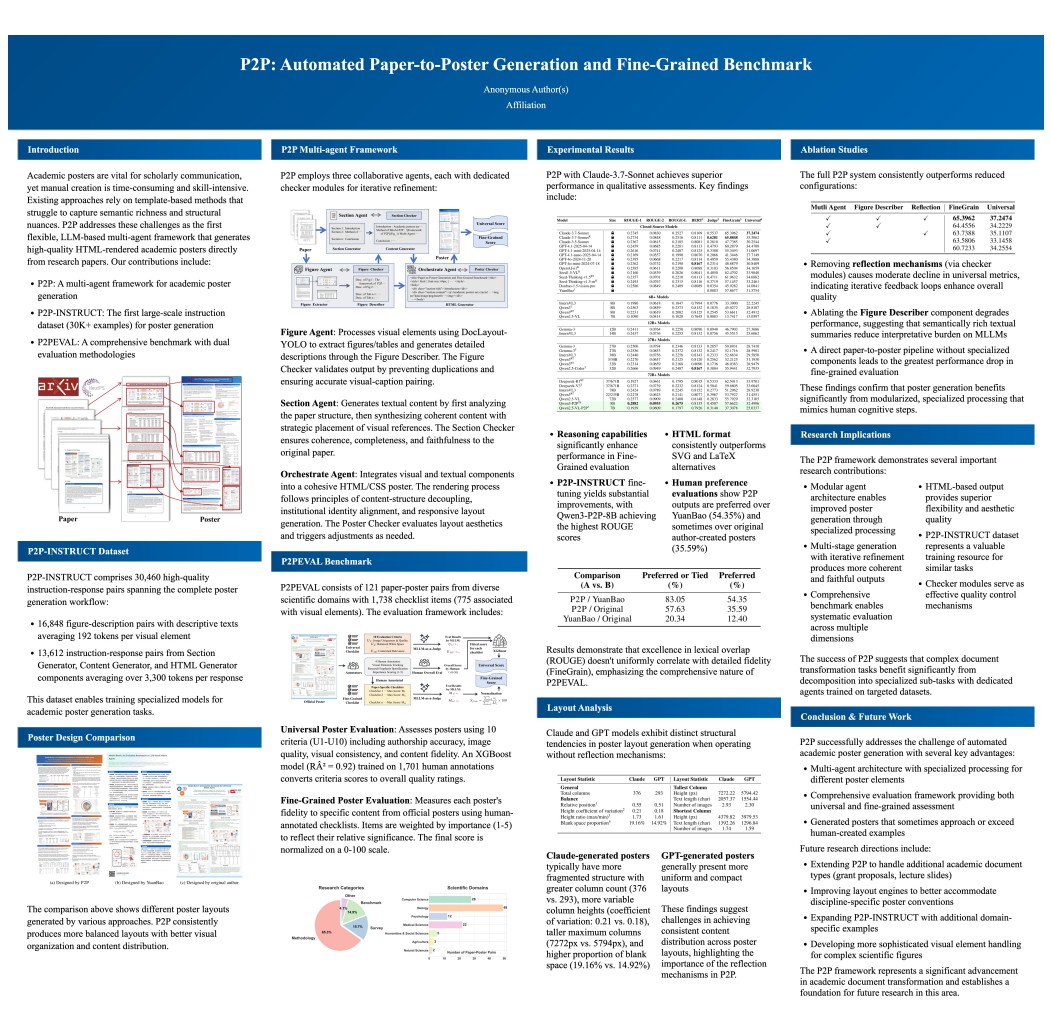

Figure 7: The poster for this paper, powered by P2P.

## P   PROMPT

---

**Section Extraction**

You are an expert in academic paper analysis.

Please analyze the paper content and identify the sections that should be included in the poster.

For each section, provide a simple description of what should be included. First, determine the type of paper. If it is a methodology research paper, focus on the method description, experimental results, and research methodology. If it is a benchmark paper, pay attention to task definitions, dataset construction, and evaluation outcomes. For survey/review papers, emphasize the significance of the field, key timelines or developmental milestones, critical theories and techniques, current challenges, and emerging trends. The above are just references; the specific section names should depend on the paper's content.

Relevant sections for comparison can be combined. There should not be too many sections. The acknowledgement and references section should not appear.

Return the result as a JSON object with section names as keys and descriptions as values.

Ensure the JSON is flat, without nested dictionaries or complex structures.

**Example Format:**
*(JSON Format Example.)*

**Paper Content:**
*(Content of Paper.)*

---

**Image Description**

You are an academic image analysis expert. Your task is to provide detailed descriptions of academic figures, diagrams, charts, or images. Describe what the figure shows, its potential purpose in an academic paper, and any key data or trends visible. The description should be concise and to the point, and should not exceed 100 words.

**Image Data:**
*(Base64 PNG Image Data.)*

---

**Text-based Poster Generation**

You are a helpful academic expert, who is specialized in generating a text-based paper poster, from given contents.

**Figure Description:**
*(Figures with Description.)*

**Paper Content:**
*(Content of Paper.)*

If the content of the poster can be described by figures, the relevant text-based content must be simplified to avoid redundancy. Important mathematical formulas can be appropriately placed to assist in understanding.

All sections should be detailed in a markdown format. Do not use headings.

---

**Image-based Poster Generation**

You are a helpful academic expert, who is specialized in generating a paper poster, from given contents and figures.

**Figure Description:**
*(Figures with Description.)*

**Text-based Poster:**

*(Text-based Poster Content.)*

**Paper Content:**
*(Content of Paper.)*

Help me inside insert figures into my poster content using my figure index as ''

figure_index starts from 0 and MUST be an integer, and don't use any other string in the figure_index.

Each figure can only be used once, and its placement should be precise and accurate.

Use pictures and tables based on their importance.

## Poster Rendering

You are a professional academic poster web page creator and your task is to generate the HTML code for a nicely laid out academic poster web page based on the object provided.

**Object Description:**

- The object contains several fields. Each field represents a section, except for the title, author and affiliation fields. The field name is the title of the section and the field value is the Markdown content of the section.
- The image in Markdown is given in the format .

**HTML Structure:**

- Only generate the HTML code inside <body>, without any other things.
- Do not use tags other than <div>, <p>, <ol>, <ul>, <li>, , , .
- Do not create sections that are not in the object.
- Place title, author and affiliation inside <div class="poster-header">. Place title inside <div class="poster-title">, author inside <div class="poster-author"> and affiliation inside <div class="poster-affiliation">.
- Place content inside <div class="poster-content">.
- Place each section inside <div class="section">. Place section title inside <div class="section-title"> and section content inside <div class="section-content">.
- Use <p> for paragraphs.
- Use <ol> and <li> for ordered lists, and <ul> and <li> for unordered lists.
- Use  for images.

**Color Specification:**

- Do not add styles other than color, background, border, box-shadow.
- Do not add styles like width, height, padding, margin, font-size, font-weight, border-radius.
- Pick at least 2 colors from the visual identity of the affiliation. If there are multiple affiliations, consider the most well-known affiliation.
- For example, Tsinghua University uses #660874 and #d93379, Beihang University uses #005bac and #003da6, Zhejiang University uses #003f88 and #b01f24. These are just examples, you must pick colors from the visual identity of the affiliation.
- Add text and background color to poster header and section title using inline style. Use gradient to make the poster more beautiful.
- The text and background color of each section title should be the same.

**Layout Specification:**

- Optionally, inside <div class="poster-content">, group sections into columns using  and <div class="poster-column" style="flex: 1">.
- You must determine the number and flex grow of columns to make the poster more balanced. If the height of one column is too large, move some sections into other columns.

- Optionally, inside <div class="section-content">, group texts and images into columns using  and <div class="section-column" style="flex: 1">.
- For example, if there are two images in two columns whose aspect ratios are 1.2 and 2 respectively, the flex grow of two columns should be 1.2 and 2 respectively, to make the columns have the same height.
- Calculate the size of each image based on columns and aspect ratios. Add comment <!– width = display_width, height = display_height –> before each image.
- Rearrange the structure and order of sections, texts and images to make the height of each column in the same group approximately the same.
- For example, if there are too many images in one section that make the height of the column too large, group the images into columns.
- DO NOT LEAVE MORE THAN 5% BLANK SPACE IN THE POSTER.

**Existing Style:**
*(Existing CSS Style.)*

**Object:**
*(Poster Object.)*

## Q  LIMITATIONS

While P2P demonstrates significant advances in automated academic poster generation, several limitations warrant acknowledgement. Our approach currently optimizes for HTML rendering, which may present compatibility challenges in academic environments where LaTeX or PowerPoint formats remain prevalent. Additionally, the system's effectiveness is constrained by the visual understanding capabilities of underlying MLLMs, occasionally resulting in misinterpretation of complex scientific visualizations or specialized notations. The multi-agent architecture, while comprehensive, introduces computational overhead that may limit accessibility for resource-constrained researchers.

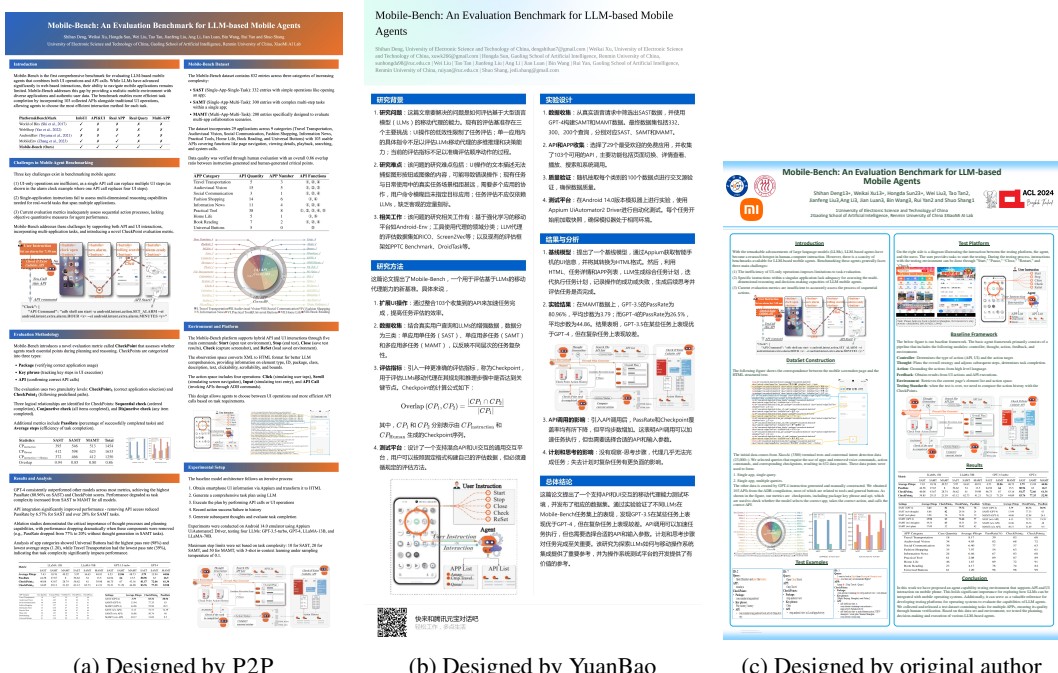

(a) Designed by P2P       (b) Designed by YuanBao       (c) Designed by original author

Figure 8: Examples of academic poster design for Deng et al. (2024).

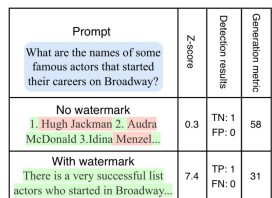
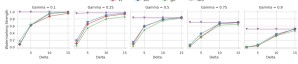
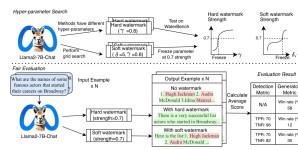
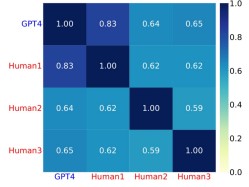
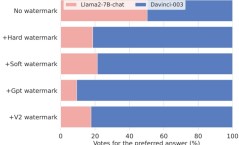

Figure 9: Examples of academic poster (Tu et al., 2023), powered by P2P.

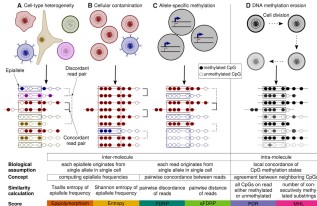
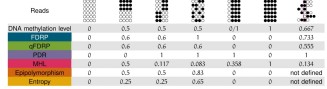
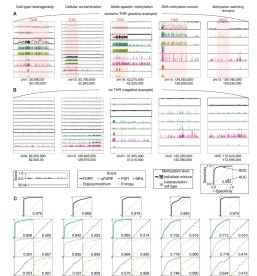
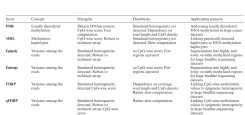
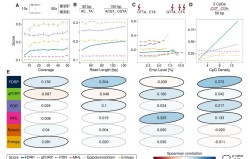
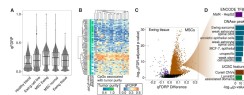

Figure 10: Examples of academic poster (Scherer et al., 2020), powered by P2P.

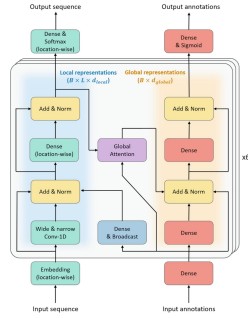
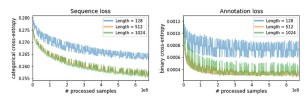
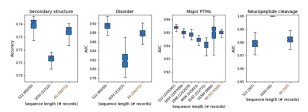
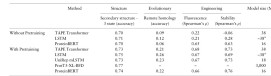
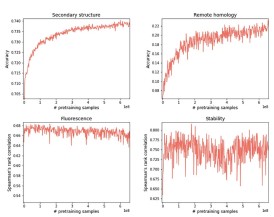
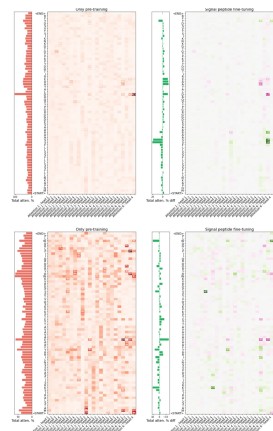

Figure 11: Examples of academic poster (Brandes et al., 2022), powered by P2P.

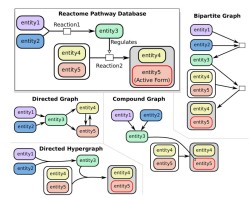
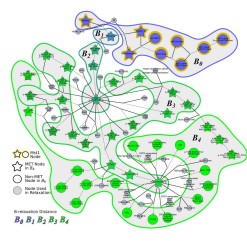
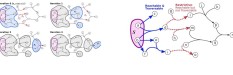
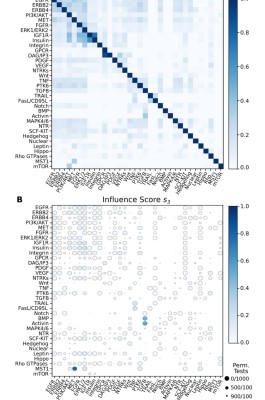
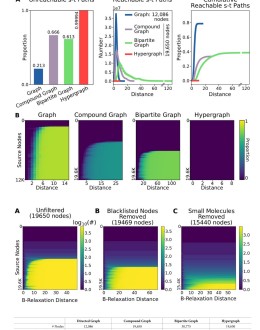
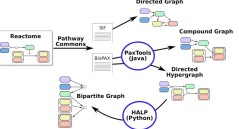
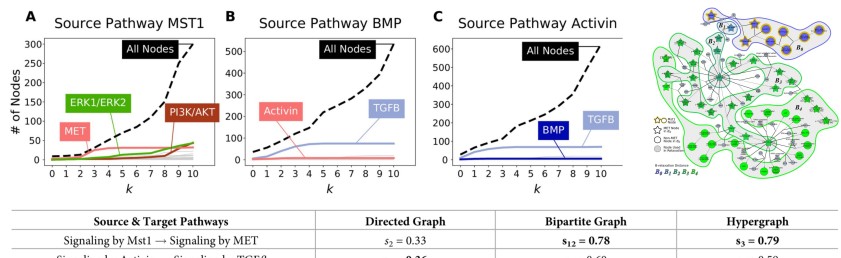

Figure 12: Examples of academic poster (Franzese et al., 2019), powered by P2P.

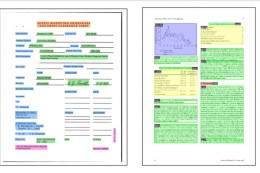
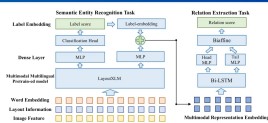
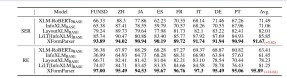
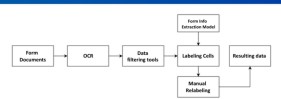
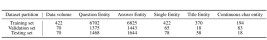
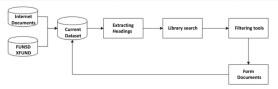
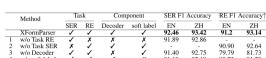

Figure 13: Examples of academic poster (Cheng et al., 2024a), powered by P2P.

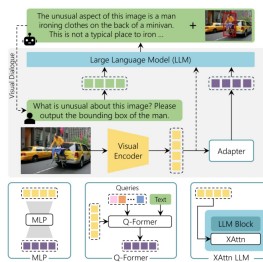
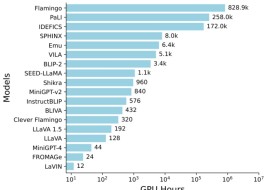

Figure 14: Examples of academic poster (Caffagni et al., 2024), powered by P2P.

# Evidence for Evolutionary and Nonevolutionary Forces Shaping the Distribution of Human Genetic Variants near Transcription Start Sites

Giovanni Scala, Ornella Affinito, Gennaro Miele, Antonella Monticelli, Sergio Cocozza

Gruppo Interdipartimentale di Bioinformatica e Biologia Computazionale, Università degli Studi di Napoli "Federico II", Naples, Italy

## Introduction

Transcription start sites (TSSs) and their surrounding regions are critical for gene regulation but are also subject to transcription-related mutagenic processes. This study investigates the genome-wide distribution of single nucleotide polymorphisms (SNPs) in the 10kb regions flanking human TSSs to understand the forces that create and maintain genetic variability in these functionally important regions.

## Methodology

We analyzed 27,487 human TSSs, categorizing them into TSSs located inside CpG islands (CGI-TSSs, 53%) and those outside (nCGI-TSSs, 47%). We classified ~2.6 million SNPs into four frequency groups based on minor allele frequency (MAF): rare (MAF$\leq 4.59 \times 10^{-4}$), mid1 ($4.59 \times 10^{-4}$ 0.01). For each TSS, the surrounding 10kb region was divided into 200 bins of 50bp each to calculate normalized mean variant frequency (BVF). We examined correlations with nucleosome positioning scores (GERP), evolutionary conservation (GERP), GC-biased gene conversion (gBGC), and variant deleteriousness (CADD).

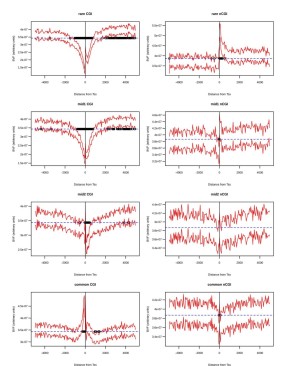

## Distribution of Variants by Frequency

Our analysis revealed that variant distribution depends on their frequency and location relative to TSSs, with distinct patterns between CGI-TSSs and nCGI-TSSs. CGI-TSSs showed a significant positional effect for all frequency classes, with a marked depression of rare variants near the TSS but a relative 1.7-fold increase in the first 200bp downstream. Conversely, nCGI-TSSs showed positional effects only for rare variants, with a smaller 1.15-fold downstream increase. Common variants showed a sharp peak near CGI-TSSs, completely absent in nCGI-TSSs, suggesting influences beyond random mutation.

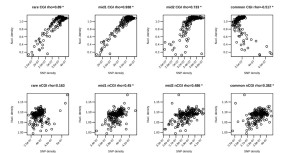

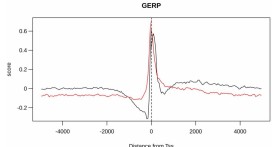

## Nucleosome Occupancy and Rare Variants

We found a strong positive correlation between nucleosome positioning and rare variant density in CGI-TSSs ($\rho=0.89$), and a weaker but significant correlation in nCGI-TSSs. CGI-TSSs exhibited a pronounced nucleosome depletion directly at the TSS, while nCGI-TSSs maintained relatively consistent nucleosome density. These findings suggest that transcription-related mutational phenomena could be linked to reduced DNA repair efficiency in nucleosome-occupied regions, as nucleosomes can limit the accessibility of repair proteins, with damage within nucleosome cores repaired at approximately 10% the rate of naked DNA.

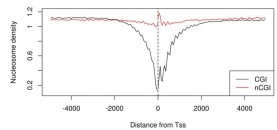

## Evolutionary Forces

Using GERP conservation scores, we identified signatures of purifying selection around TSSs. For nCGI-TSSs, we found a strong positive correlation ($\rho=0.725$) between GERP scores and BVF-delta (difference between rare and common variant frequencies). For CGI-TSSs, we observed a complex pattern with a strong positive correlation ($\rho=0.774$) between GERP and BVF-delta in regions >700bp from TSSs, suggesting purifying selection preserves functional regions by preventing deleterious mutations from reaching common frequencies. These evolutionary constraints are stronger in CGI-TSSs, consistent with their association with housekeeping genes that require stringent conservation.

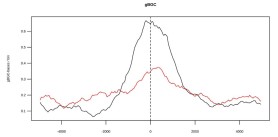

## Non-evolutionary Forces

We identified GC-biased gene conversion (gBGC) as a significant non-evolutionary force affecting allele frequencies near TSSs. In CGI-TSSs, we found a strong negative correlation ($\rho=-0.734$) between gBGC scores and BVF-delta within the inner 700bp region around TSSs. This indicates that gBGC competes with purifying selection, particularly in CpG-rich regions, by preferentially resolving GC/AT heterozygotes to GC/GC homozygotes during gene conversion, thereby increasing the frequency of variants that might otherwise be selected against.

## Variant Deleteriousness

Analysis of CADD scores revealed that variants closer to TSSs are potentially more deleterious than those more distant, with deleteriousness increasing toward the TSS from both sides. Rare variants consistently showed higher deleteriousness scores than common variants across all positions, supporting the notion that purifying selection prevents deleterious mutations from reaching high frequencies. Furthermore, variants in CGI-TSSs exhibited significantly higher deleteriousness scores than those in nCGI-TSSs within approximately 1300bp of the TSS, highlighting the functional importance of CGI regions.

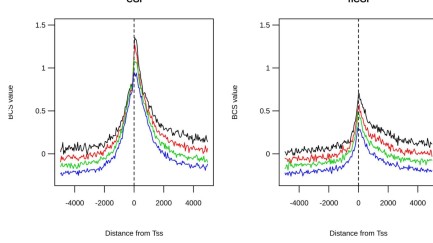

## Conclusions

This study provides a detailed view of how human genetic variants are distributed around TSSs, revealing that both evolutionary (purifying selection) and non-evolutionary (gBGC) forces shape genetic variability in these critical regulatory regions. Rare variants show strong correlations with nucleosome positioning, suggesting transcription-related mutagenic processes influence their distribution. The competing effects of purifying selection and gBGC create distinctive frequency patterns, particularly in CGI-TSSs, while the higher deleteriousness of variants near TSSs underscores the functional importance of these regions. These findings enhance our understanding of the complex interplay between mutational processes and selective forces in shaping human genomic diversity.

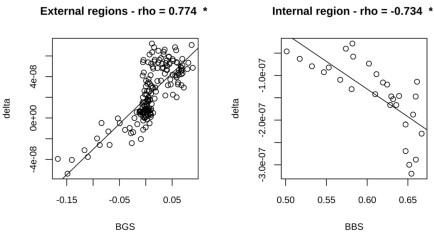

Figure 15: Examples of academic poster (Scala et al., 2014), powered by P2P.

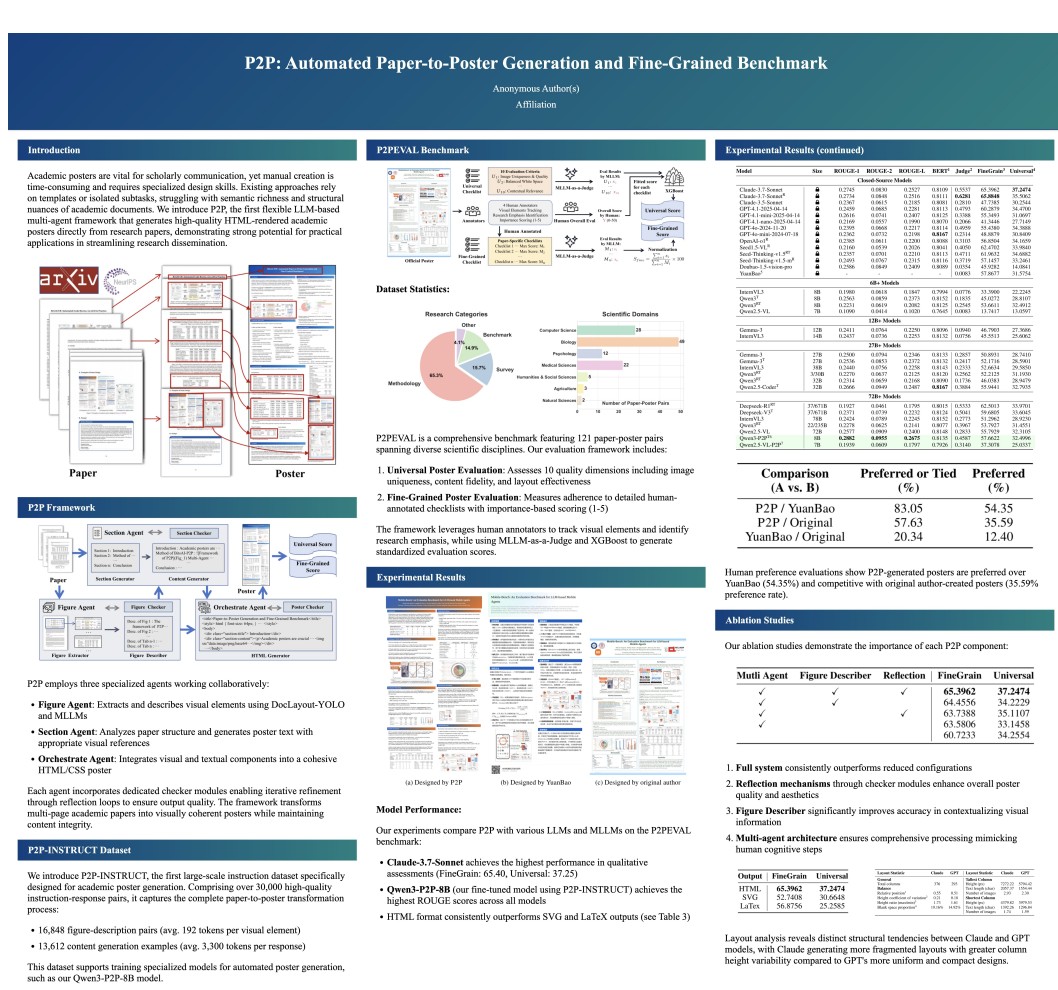

Figure 16: The horizontal poster for this paper, powered by P2P.

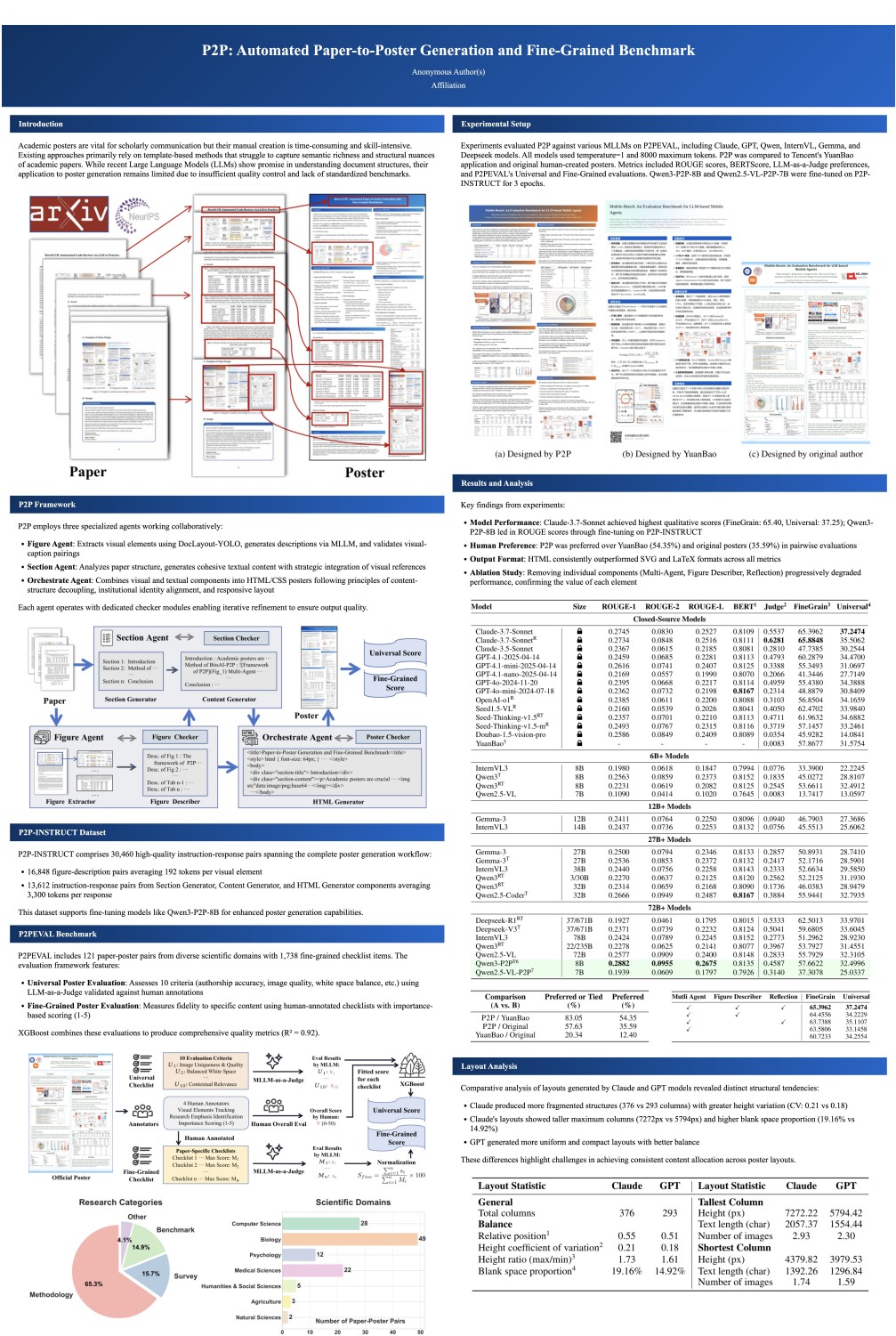

Figure 17: The vertical poster for this paper, powered by P2P.

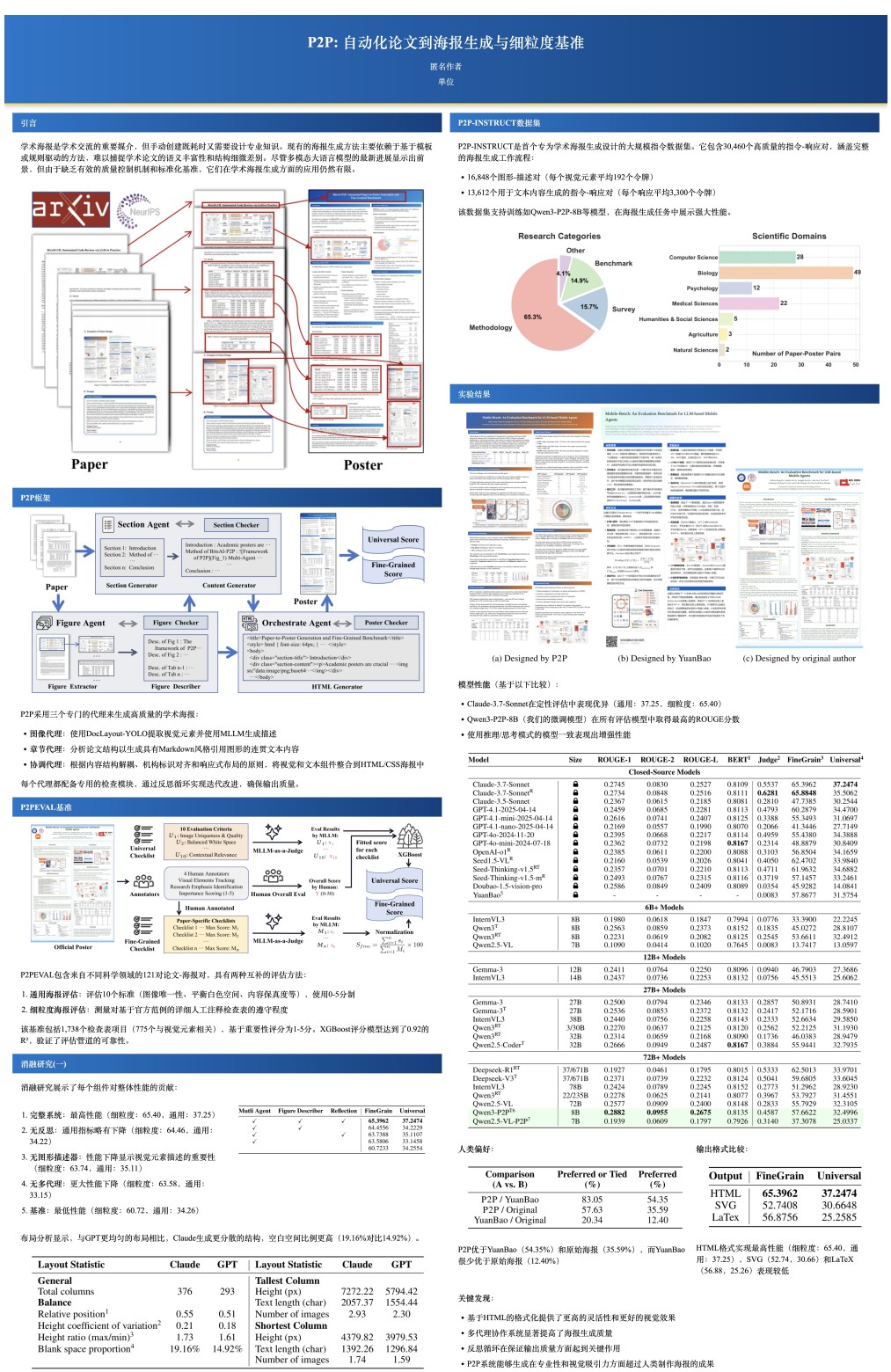

Figure 18: The poster for this paper in another language (Chinese), powered by P2P.

