# OpenReview forum: "P2P: Automated Paper-to-Poster Generation and Fine-Grained Benchmark"
_ICLR.cc/2026/Conference — ICLR 2026 Poster_

### Official Review · Reviewer_4iW7 · 2025-10-29

**Soundness:** 3
**Presentation:** 3
**Contribution:** 3
**Rating:** 6
**Confidence:** 3

**Summary:**

This paper presents a comprehensive and well-executed study on the automated generation of academic posters from research papers. The work is timely, addressing a clear need in the academic community. The proposed P2P framework, the P2PINSTRUCT dataset, and the P2PEVAL benchmark constitute a significant contribution to the field of document AI and scientific communication. The paper is generally well-written and the experimental evaluation is extensive. However, some aspects of the methodology and presentation could be strengthened to improve clarity and impact.

**Strengths:**

1.   This is one of the first works to systematically tackle end-to-end academic poster generation using an LLM-based multi-agent framework. The problem is relevant and underexplored. The creation of P2PINSTRUCT and P2PEVAL fills a critical gap, providing essential resources for future research in this domain.
2.   The dual-evaluation framework of P2PEVAL is a major strength. The argument for decoupling objective fidelity (Fine-Grained) from subjective quality (Universal) is well-motivated. The use of human-annotated checklists and a learned model for holistic scoring is sophisticated and convincing. The high R² score (0.92) for the Universal Evaluation model adds credibility.
3.  The paper evaluates a vast number of models (35), including both closed-source and open-source LLMs/MLLMs. The ablation studies effectively demonstrate the contribution of each component (multi-agent, figure describer, reflection). The human preference evaluation and analysis of output formats (HTML vs. SVG/LaTeX) provide valuable practical insights.
4.   The authors commit to releasing code, datasets (P2PINSTRUCT, P2PEVAL), and detailed prompts, which will greatly facilitate reproducibility and future work. The ethical considerations and resource reporting are appropriate.

**Weaknesses:**

1.  While the paper extensively compares different LLM backbones, it lacks a direct comparison to prior dedicated poster-generation systems (e.g., Qiang et al., 2019; Jaisankar et al., 2024 - Postdoc). A qualitative or quantitative comparison against such established, non-LLM-based methods would better situate the performance of P2P and highlight its advancements. The current comparison feels more like a model benchmark than a system-level comparison.
2.  The relationship between the P2P *framework* and the specific LLMs used within it (e.g., Claude-3.7-Sonnet) needs to be clarified. Is P2P best viewed as a prompting/orchestration strategy that can be deployed on top of any powerful LLM? The results in Table 1 show "P2P" achieving high scores, but this seems to be the result of using the P2P *pipeline* with Claude-3.7-Sonnet as the core LLM. The fine-tuned models (Qwen3-P2P) are a separate contribution. This distinction should be made more explicit to avoid confusion.
3.   The paper convincingly argues for HTML's advantages (flexibility, interactivity). However, the limitation regarding the prevalence of LaTeX/PPT in academia is buried in the appendix (Section M). This is a significant practical constraint that deserves more prominent discussion in the main text. How might P2P be adapted or extended to output these more traditional formats? Are there plans for a PDF/LaTeX export feature?
4.   The multi-agent framework with iterative reflection loops is computationally expensive. While resource consumption is mentioned in Appendix G, a more detailed discussion in the main limitations section about the inference-time cost and latency of the full P2P system would be valuable for potential users.

**Questions:**

1.  The P2PINSTRUCT dataset is generated by the P2P framework itself. Could there be a risk of a "closed loop" where the dataset inherits and potentially amplifies any systematic biases or error modes of the P2P pipeline? How was the quality of this synthetic data validated?
2.  The Universal Evaluation uses an XGBoost model to map LLM-generated criterion scores to a human holistic score. Was there any exploration into using a fine-tuned LLM as the holistic judge directly? What was the rationale for choosing a simpler model like XGBoost over an LLM for this final aggregation step? Furthermore, how does the performance of this XGBoost model generalize to posters generated by future, potentially very different, methods not seen during its training?
3.  The checker modules are a critical component for iterative refinement. Could you provide more concrete examples of typical failure cases that the checkers identified and how the reflection loops successfully corrected them? Conversely, are there systematic errors or types of source paper content that the checkers consistently fail to catch, leading to persistent issues in the final poster?
4.  The P2PEVAL benchmark is constructed from 121 paper-poster pairs. Could you provide more detail on the stratification and diversity of this benchmark? For instance, how are papers from different scientific fields (e.g., computer science, NLP, CV) and types (e.g., methodological, theoretical, survey) distributed? Is there a risk that the benchmark over-represents certain fields, and how might this affect the fair evaluation of a model's general capabilities?

---

> ### Author Response · Authors · 2025-11-22
> **Response to Reviewer 4iW7 (1/2)**
>
> We sincerely thank the reviewer for their thorough and insightful feedback. The reviewer's suggestions are excellent and will significantly strengthen the paper. We address each point below and will incorporate these clarifications into the rebuttal version.
>
> ---
>
> ## W1: Comparison to Prior Dedicated Systems
> This is an excellent point. We agree that situating `P2P` against prior work[1,2], both of which we cite, is crucial. Unfortunately, after an extensive search, we were unable to find official or unofficial implementations of these systems for a direct quantitative comparison. However, based on the examples provided in their papers, we believe `P2P`'s outputs demonstrate a significant improvement in quality and layout flexibility. To provide a concrete comparison against a contemporary system, we included results from Tencent's YuanBao in Table 1, which further highlights the strong performance of our framework.
>
> [1] PostDoc: Generating Poster from a Long Multimodal Document Using Deep Submodular Optimization. Vijay Jaisankar, Sambaran Bandyopadhyay, Kalp Vyas, Varre Chaitanya, Shwetha Somasundaram. 2024.
>
> [2] Learning to Generate Posters of Scientific Papers by Probabilistic Graphical Models. Yu-ting Qiang, Yanwei Fu, Xiao Yu, Yanwen Guo, Zhi-Hua Zhou, Leonid Sigal. 2019.
>
> ---
>
> ## W2: Clarifying the `P2P` Framework vs. the LLM Backbone
> Thank you for highlighting this ambiguity. You are correct: `P2P` is a **model-agnostic orchestration framework** that can be instantiated with various LLMs. The results in Table 1 demonstrate the performance of this framework when using different models as its core components(Figure Describer, Section Generator, Content Generator, HTML Generator). `Claude-3.7-Sonnet` achieved high scores, validating its strong capabilities, but the framework itself is versatile. The results from YuanBao and other models further illustrate how `P2P` serves as a robust pipeline for this task.
>
> ---
>
> ## W3: Prominence of the HTML Output Limitation
> This is a significant practical limitation that deserves more prominent discussion. Due to page limits, we initially placed it in the appendix, but we will revise the paper's layout to include it in the main text.
>
> While HTML offers benefits, we acknowledge that LaTeX/PPT are more convenient for users unfamiliar with web technologies. As shown in Table 3, modifying the Orchestrate Agent to generate LaTeX introduces a performance overhead, particularly in visual-textual arrangement. We hypothesize this is due to biases in current LLMs' training data; a Google Scholar search (as of Nov 22, 2025) for "LLM generate HTML/PPT/LaTeX" yields 80,200, 5,310, and 7,720 results, respectively, suggesting greater community focus on HTML generation. For future work, we note that HTML can be easily exported to PDF. While direct LaTeX/PPT generation requires further research, emerging tools (like Doubao's visual HTML editor) could bridge this usability gap.
>
> ---
>
> ## W4: Computational Cost of the P2P
> We agree that a more detailed discussion of the inference cost is valuable. We will add a new subsection in the appendix detailing the costs for representative closed-source (GPT-4.1) and open-source (Qwen-2.5-VL-7B) models.
>
> With multi-threading and parallel optimizations, generating a poster for an average paper takes **~55 seconds for Qwen-2.5-VL-7B** and **~209 seconds for GPT-4.1**.
> In terms of token costs, for an average paper (17,104 text tokens, 7.25 images, from Table 6), the approximate cost is as follows (these are estimates due to the checker/reflection loops):
> - **GPT-4.1**: For a single poster, the total token consumption is approximately 88,031 input tokens and 9,577 output tokens. This includes an estimated retry rate of 1.05x for the Section Agent and 2.32x for the Orchestrate Agent. At OpenRouter's pricing ( 2 dollars/M for input,  8 dollars/M for output), the calculation is: (88,031 / 1,000,000 \*  2) + (9,577 / 1,000,000 \* 8) ≈ \$ 0.25.
> - **Qwen-2.5-VL-7B**: The same process requires approximately 107,993 input tokens and 8,286 output tokens, with different retry rates (1.03x for Section, 3.47x for Orchestrate). At its pricing (0.20 dollars /M for both input/output), the calculation is: (107,993 / 1,000,000 \*  0.20) + (8,286 / 1,000,000 \*  0.20) ≈ \$ 0.02.
>
> We will add this detailed breakdown to the appendix to provide full transparency on the cost-quality trade-off.

---

> > ### Author Response · Authors · 2025-11-22
> > **Response to Reviewer 4iW7 (2/2)**
> >
> > ## Q1: Risk of "Closed Loop" Bias in P2Pinstruct
> >
> > This is an insightful question about synthetic data. We took several steps to mitigate this risk and validate data quality. Our `P2Pinstruct` is a multi-modal instruction dataset where each step in the generation process is informed by the ground truth human-created poster. For example, the HTML generator is prompted with the Markdown content but also receives the original poster image as a layout reference. This disassembles the human creation process into `P2P`'s stages, anchoring the synthetic data to human design principles.
> >
> > And the dataset was constructed from the SciPostLayout training set (which does not overlap with test set) due to its CC-BY license and diverse subject matter.
> >
> > We validated the quality of this dataset in three ways:
> > 1.  **Built-in Quality Control:** The data was generated using the full `P2P` with checker-reflection loops enabled, providing automated quality control.
> > 2. **Manual Verification**: We also manually checked a random sample of 20 examples from each generation stage to confirm quality.
> > 3.  **Downstream Performance:** As shown in Table 1, models fine-tuned on `P2Pinstruct` (e.g., `Qwen3-P2P`) significantly outperforming their base versions. This empirical improvement is evidence of the dataset's effectiveness.
> >
> > ---
> >
> > ## Q2: Rationale for XGBoost in Universal Evaluation
> >
> > This is an excellent question. We explored using an LLM as the holistic judge directly, but chose XGBoost for its stability and reliability in this specific regression task.
> >
> > Our two-step approach combines the strengths of both paradigms:
> > 1.  **Stable Feature Extraction (LLM):** We use the LLM for what it does best: scoring discrete, well-defined criteria (U1-U10). This is far more reliable than asking for a single, noisy holistic score.
> > 2.  **Learning Human Preferences (XGBoost):** Humans weigh these 10 criteria in complex, non-linear ways. XGBoost excels at learning this mapping from our 1,701 human preference ratings.
> >
> > Follow Appendix D.1, we conducted experiments based on Qwen-2.5-VL-32B to validate this choice:
> >
> > | Method| R² | KL-Divergence| JS-Divergence |
> > | - | - | - | - |
> > | **Direct LLM Scoring** (End-to-End, prompted the LLM to directly output a holistic Universal Score)   | 0.27           | 1.60|0.62|
> > | **LLM as Regressor (Replacing XGBoost)** (provided the LLM with the 10 criteria scores and asked it to produce the Universal Score) | 0.51| 0.34| 0.22|
> > | **Fine-Tuned LLM as an Aggregator** (use same dataset for XGBoost)| 0.70| **0.06** | 0.14|
> > | **Our MLLM-Featurizer + XGBoost** | **0.92** | 0.11| **0.02** |
> >
> > This large performance gap validates our design. Our findings align with other work [3,4] suggesting that for certain regression tasks, well-established models like XGBoost can outperform LLMs. We will add these new results to the appendix.
> >
> > [3] Predicting Learning Performance with Large Language Models: A Study in Adult Literacy. Liang Zhang, Jionghao Lin, Conrad Borchers, John Sabatini, John Hollander, Meng Cao, Xiangen Hu. 2024.
> >
> > [4] Regression with Large Language Models for Materials and Molecular Property Prediction. Ryan Jacobs, Maciej P. Polak, Lane E. Schultz, Hamed Mahdavi, Vasant Honavar, Dane Morgan. 2024.
> >
> > ---
> >
> > ## Q3: Concrete Examples of the Checker-Reflection Loop
> > We agree that a concrete example would be very helpful. We will add a visual example in the appendix. For instance, where the *Poster Checker* detects excessive blank space, it triggers a reflection loop. Then *Orchestrate Agent* is re-prompted to re-balance the columns, successfully correcting the layout.
> >
> > We will also note that in rare cases where a model repeatedly fails to satisfy a check (e.g., after 5 retries), the P2P can escalate the issue by re-triggering an earlier agent (e.g., the *Section Agent*) to generate entirely new content.
> >
> > ---
> >
> > ## Q4: Diversity of the `P2Peval` Benchmark
> > We ensured `P2Peval` is diverse. All papers were manually classified by our annotators, who have relevant research experience. To ensure the benchmark's universality, we made a special effort to find posters from non-computer science fields. As shown in Figure 4 and Appendix C, the papers are sourced from Computer Science conferences and the multi-disciplinary SciPostLayout (covering Biology, Medicine, Psychology, etc.). We avoided using papers from sources like ICLR after consulting with legal professionals, as their copyrights would require author-by-author permissions.
> >
> > Crucially, the performance breakdown in Tables 7 & 8 shows the benchmark is well-balanced. For example, our best model achieves the highest Fine-Grained score in Psychology and the highest Universal score in Computer Science. This indicates the benchmark does not unfairly favor one domain and provides a robust testbed for evaluating general poster generation capabilities.
> >
> > ---
> >
> > We thank the reviewer again for their constructive feedback. We are confident that these changes will significantly improve our paper.

---

> > > ### Author Response · Authors · 2025-11-28
> > > **A Friendly Reminder for Reviewer 4iW7**
> > >
> > > Dear Reviewer 4iW7,
> > >
> > > Thank you again for your thoughtful and constructive comments. We wanted to kindly let you know that we have already provided detailed responses to your questions regarding **the practical computational cost and latency of our pipeline, and the rationale for our choice of XGBoost** over the past six days. We have also incorporated these clarifications and analyses into the updated version of the paper.
> > >
> > >
> > > **We hope our responses and the revised manuscript have thoroughly addressed your questions, and we look forward to any further discussion or feedback.**

---

### Official Review · Reviewer_HwQu · 2025-10-30

**Soundness:** 3
**Presentation:** 3
**Contribution:** 2
**Rating:** 4
**Confidence:** 3

**Summary:**

This paper addresses the time-consuming task of creating academic posters from research papers. The authors present a comprehensive, three-part contribution:

1. **P2P:** A flexible, LLM-based multi-agent framework that generates HTML-rendered posters. It employs three specialized agents (Figure Agent, Section Agent, Orchestrate Agent) and a novel "checker-reflection" mechanism to iteratively refine the output.
2. **P2PEVAL:** A new, comprehensive benchmark for evaluating generated posters. Its core innovation is a dual-evaluation methodology, assessing both **objective fidelity** (via a "Fine-Grained" score based on 1738 human-annotated checklist items) and **subjective quality** (via a "Universal" score from an XGBoost model trained to emulate human aesthetic preferences).
3. **P2PINSTRUCT:** The first large-scale (30,000+ examples) instruction dataset for the paper-to-poster task, derived from the intermediate outputs of the P2P pipeline.

Experiments show the P2P system, particularly when powered by Claude-3.7-Sonnet, outperforms 35 other models. Notably, human evaluations found the P2P-generated posters to be competitive with, and in many cases preferred to, the original posters created by the authors themselves.

**Strengths:**

1. The authors have built an end-to-end solution, from the generation framework (P2P) to the large-scale data for training (P2PINSTRUCT) and a novel benchmark for evaluation (P2PEVAL). This is a significant and impressive amount of work.
2. The authors conducted comprehensive experiments, including evaluation on 35 models and a human preference study. The P2P framework is shown to be effective via an ablation study.

**Weaknesses:**

1. There is a lack of analysis for the effectiveness of the P2PINSTRUCT dataset.
2. The methodology of this metric measures fidelity to a specific human instance, which itself may be suboptimal (as shown by human preference research). The authors should discuss this tension between "imitation" and "fidelity" more openly.
3. The two-step, MLLM-Featurizer-plus-XGBoost-Regressor methodology for the universal score is not justified over simpler, more direct MLLM-as-a-Judge approaches. This adds a "Rube Goldberg"-like complexity without a clear benefit.

**Questions:**

1. In Table 1, models fine-tuned on P2PINSTRUCT (like Qwen3-P2P) achieve the absolute best scores on ROUGE and BERTScore. However, the improvement on the paper's own flagship metrics, Fine-Grained and Universal, is limited. Does this mean there is a bias in the way this dataset was constructed?
2. Could the authors more clearly state in the paper that the purpose of the "Fine-Grained" metric is to measure fidelity to "the key content selected by the authors" rather than imitation of "the layout design of the authors"?
3. The "checker-reflection" mechanism is interesting. Can you provide a specific example? For example, when the Poster Checker detects a "spacing imbalance," what kind of "reflection" instructions does it send to the Orchestrate Agent to correct it? How many iterations does this process typically take?
4. Could the authors provide an ablation experiment or justification to demonstrate that the MLLM-XGBoost process is superior to the simpler, end-to-end MLLM-as-a-Judge baseline?

---

> ### Author Response · Authors · 2025-11-22
> **Response to Reviewer HwQu (1/2)**
>
> Thanks for your constructive feedback and for acknowledging the significant amount of our work. We appreciate your thoughtful comments, which have helped us clarify our contributions. We address your points below.
>
> ---
>
> ### W1 & Q1: On the Effectiveness and Perceived Bias of `P2Pinstruct`
> We agree that a deeper analysis of `P2Pinstruct`'s effectiveness is crucial.
>
> **1. Clarifying the Purpose and Effectiveness of `P2Pinstruct`:** The `P2Pinstruct` dataset is derived from the intermediate inputs and outputs of our multi-agent pipeline. Its primary goal is to train models on the foundational subtasks of poster generation: summarizing sections, describing figures, and structuring content with Markdown/HTML. These are text-heavy tasks, which is precisely what ROUGE and BERTScore measure. The strong improvements on these metrics therefore confirm that `P2PINSTRUCT` is highly effective for its designed purpose. We will add a new Appendix section with detail of the P2Pinstruct Dataset.
>
> **2. Quantifying the Gains on Fidelity and Quality:** While the gains on Fine-Grained and Universal scores are more modest than on lexical metrics, they are consistent and significant, demonstrating that instruction-tuning also improves higher-level content fidelity and design quality. The improvements are not "limited," but represent a stable positive impact:
> *   **Qwen3-P2P (8B)** vs. base: Fine-Grained +**12.64**, Universal +**3.69**
> *   **Qwen2.5-VL-P2P (7B)** vs. base: Fine-Grained +**23.57**, Universal +**11.97**
> *   **InternVL3-P2P (8B)** vs. base: Fine-Grained +**18.58**, Universal +**9.40**
>
> These substantial gains prove the dataset's value. The smaller delta compared to ROUGE is because Fine-Grained and Universal scores also depend on complex skills like visual grounding and layout reasoning, which are more inherent to the backbone model's capabilities. Open-source models still have a gap compared with closed-source models. This is not evidence of dataset bias, but an indicator of which skills are most improved by our current instruction set.
>
> ---
>
> ### W2 & Q2: Clarifying the Purpose of the Fine-Grained Metric
> This is an excellent point. We will clarify the distinction between fidelity and imitation.
> The purpose of our **Fine-Grained** metric is to measure **fidelity to the key content selected by the original author**, not to imitate their specific layout or design. We treat the original poster as a "ground truth" reference for what the author or a domain expert, deemed the most critical figures, conclusions, and structural information. Our checklist-based evaluation (Section 3.1) formalizes this by asking verifiable questions like, "Does the poster include Figure 3, which shows performance on dataset X?" or "Does it state the main conclusion about Y?"
>
> Our `P2Peval` actively **decouples content fidelity from layout design**, allowing us to focus on accurately representing the author's scientific message. The fact that our generated posters are often preferred to the originals further supports this. Our human preference study (Table 2) shows that `P2P`-generated posters were judged as **superior or equal to the original author's in 57.63% of cases**, and strictly preferred in 35.59%. This demonstrates that our system can produce outputs that are not only faithful in content but often superior in holistic quality.

---

> > ### Author Response · Authors · 2025-11-22
> > **Response to Reviewer HwQu (2/2)**
> >
> > ## W3 & Q4: Justifying the MLLM-Featurizer + XGBoost Methodology
> >
> > We appreciate the "Rube Goldberg" analogy and agree that this complexity requires strong justification. Our two-step approach is a principled design choice to create a more robust and human-aligned evaluation.
> >
> > **1. The Problem with Direct LLM Judging:**
> > Directly asking an LLM for a single holistic score (e.g., 0–50) is often noisy, inconsistent, and lacks interpretability. Our solution combines the strengths of different models:
> > *   **Stable Feature Extraction (LLM):** We decompose the abstract concept of "quality" into 10 concrete, well-defined criteria (U1-U10). LLMs are far more reliable at scoring these specific attributes than a single abstract score, yielding a stable feature score.
> > *   **Learning Human Preferences (XGBoost):** Humans weigh these 10 criteria non-linearly. For example, a poor information flow (U9) is a critical flaw, while slightly unbalanced white space (U3) is a minor issue. XGBoost excels at learning these complex, non-linear relationships from data. We trained it on 1,701 human preference scores to create a "human preference function."
> >
> > **2. New Experiment (as suggested):**
> > To empirically validate this, we conducted the experiment you suggested. We compared our two-step method against baselines using Qwen2.5-VL-32B:
> > | Method|R²| KL-Divergence  | JS-Divergence  |
> > | - | - | - | - |
> > | **Direct LLM Scoring**(End-to-End, prompted the LLM to directly output a holistic Universal Score) | 0.27| 1.60| 0.62 |
> > | **LLM as Regressor (Replacing XGBoost)**(provided the LLM with the 10 criteria scores and asked it to produce the Universal Score) | 0.51| 0.34 | 0.22  |
> > | **Fine-Tuned LLM as an Aggregator**(use same dataset for XGBoost)| 0.70           | **0.06** | 0.14  |
> > | **Our MLLM-Featurizer + XGBoost**|**0.92** | 0.11 | **0.02** |
> >
> > This large performance gap validates our design. While a fine-tuned LLM can improve, it does not match the predictive accuracy of XGBoost, which remains a superior and more computationally efficient tool for this regression task, a finding supported by other work [1, 2].
> >
> > [1] Predicting Learning Performance with Large Language Models: A Study in Adult Literacy. Liang Zhang, Jionghao Lin, Conrad Borchers, John Sabatini, John Hollander, Meng Cao, Xiangen Hu. 2024.
> >
> > [2] Regression with Large Language Models for Materials and Molecular Property Prediction. Ryan Jacobs, Maciej P. Polak, Lane E. Schultz, Hamed Mahdavi, Vasant Honavar, Dane Morgan. 2024.
> >
> > ---
> >
> > ## Q3: A Concrete Example of the Checker-Reflection Mechanism
> >
> > This is a great question. The **Poster Checker** is **rule-based and programmatic** for efficiency and deterministic evaluation. It parses the generated HTML to check for structural and aesthetic issues. We will add a visual example in the appendix.
> > *   **Example Trigger:** A key rule evaluates layout balance. If the checker calculates that the **proportion of blank space exceeds a 10% threshold**, or if column heights are severely imbalanced (as quantified by the metrics in Table 5), the reflection loop is triggered.
> > *   **Reflection Instruction:** The checker feeds these layout metrics (e.g., `"whitespace_ratio": 0.15`, `"column_heights_px": [1200, 1800, 900]`) and previous HTML code back to the Orchestrate Agent.
> > *   **Iterations:** We have measured the average number of retries on `P2PEVAL`. For GPT-4.1, it is 2.32 iterations, and for the smaller Qwen-2.5-VL-7B, it is 3.47 iterations, showing the process is efficient.
> >
> > ---
> >
> > We thank you again for your valuable feedback. We are confident that these clarifications and new analyses will strengthen the paper and more clearly articulate the novelty and rigor of our work.

---

> > > ### Author Response · Authors · 2025-11-28
> > > **A Friendly Reminder for Reviewer HwQu**
> > >
> > > Dear Reviewer HwQu,
> > >
> > > Thank you again for your thoughtful and constructive comments. We wanted to kindly let you know that we have already provided detailed responses to your questions regarding **the effectiveness of the P2Pinstruct dataset, the justification for Universal Score's methodology, and the example of checker-reflection mechanisms** over the past six days. We have also incorporated these new analyses into the updated version of the paper.
> > >
> > >
> > > **We hope our responses and the revised manuscript have thoroughly addressed your questions, and we look forward to any further discussion or feedback.**

---

### Official Review · Reviewer_Fp4t · 2025-10-31

**Soundness:** 3
**Presentation:** 3
**Contribution:** 4
**Rating:** 6
**Confidence:** 5

**Summary:**

This paper introduces a foundational ecosystem for automated academic poster generation, addressing key limitations in prior work, such as the failure to capture semantic richness and the lack of standardized benchmarks. The work is built on three main contributions:

1.  **P2P**: A flexible, LLM-based multi-agent framework designed to generate high-quality, HTML-rendered academic posters directly from research papers.
2.  **P2PINSTRUCT**: The first large-scale instruction dataset (30,000+ examples) specifically tailored for the paper-to-poster generation task.
3.  **P2PEVAL**: A comprehensive benchmark featuring 1738 checklist items and a novel dual evaluation methodology (Fine-Grained and Universal) to assess poster quality.

Unlike prior methods that often struggle with semantic nuances and lack systematic evaluation, this paper offers a complete system from generation to evaluation, providing a principled blueprint for assessing complex, creative AI-generated artifacts.

**Strengths:**

1.  **Innovative P2P Framework**
    The P2P multi-agent architecture is a novel contribution to complex document transformation tasks. The inclusion of a **checker-reflection mechanism** is particularly strong, as it mimics the human design process of drafting and revision. This iterative approach helps ensure both scientific accuracy and structural integrity in the final output.

2.  **Valuable Dataset (P2PINSTRUCT)**
    The paper introduces P2PINSTRUCT, the first large-scale (30K+) instruction dataset specifically designed to train models for the complete paper-to-poster workflow. This resource could be highly valuable for the community and spur further research in this domain.

3.  **Comprehensive Benchmark (P2PEVAL)**
    The authors have clearly put significant effort into creating P2PEVAL, a new and comprehensive benchmark (1738 checklist items, 121 pairs). The **dual-evaluation framework** is a key strength, thoughtfully separating objective fidelity (via human-annotated checklists) from subjective quality (via a predictive scoring model). This provides a robust and multifaceted analysis of poster quality.

**Weaknesses:**

### 1. Lack of Detail on P2P Checker Mechanisms
The paper states, "Each agent operates in conjunction with a dedicated checker module that triggers a reflection loop if its output fails to meet quality standards". However, the manuscript provides no concrete details on how these critical checker modules are implemented.
* What is the core component of each checker? Is it an LLM-as-a-judge, a set of programmatic rules, or a trained classifier?
* **Figure Checker**: The paper mentions "an initial confidence threshold". What is this threshold value, and how was it determined?
* **Section Checker**: This checker reportedly evaluates four complex metrics: coherence, completeness, faithfulness, and correct referencing. How are these metrics automatically and objectively measured?
* **Poster Checker**: How is this checker implemented to evaluate "layout aesthetics and structural integrity"?

### 2. Missing Implementation Details for P2P Agents
The paper describes the multi-agent architecture but omits all implementation details about the agents themselves.
* What specific models are used for the main agents ($\mathcal{A}_{Fig}$, $\mathcal{A}_{Sec}$, $\mathcal{A}_{Orch}$)?
* What models are used for their generative sub-components (e.g., Figure Describer, Section Generator, Content Generator, HTML Generator)? This is crucial for reproducibility.

### 3. Contradictory Description of Fine-Grained Poster Evaluation
I am very confused about how the Fine-Grained Poster Evaluation is actually performed, as the paper seems to contradict itself.
* In one section, the paper states it uses "LLM-as-a-Judge... to score generated posters against detailed, human-annotated checklists".
* However, in Section 3.2.1, it explicitly states, "This is a **deterministic scoring methodology, not a trained model**".
* These two statements are mutually exclusive. This core evaluation method must be clarified.

**Questions:**

### 1. Unclear Notations
Some technical notations are ambiguous and hinder a clear understanding of the model's inputs and outputs.
* L134: "$v_i$ denotes the visual". Does "visual" refer to the raw image file (e.g., a PNG) or a high-dimensional feature vector extracted from the image?
* L142: "a detailed structural schema ($S$)". What is the data type of $S$? Is it a textual representation of the poster's structure (e.g., a JSON object) or a numerical feature vector?

### 2. Lack of Detail on the P2PINSTRUCT Dataset
The paper introduces P2PINSTRUCT as a "high-quality" large-scale dataset, but more details are needed to fully assess its scope and quality. Section 2.2 mentions it's derived from the P2P framework's intermediate outputs, including prompting Claude for figure descriptions. The authors should provide more details on:

* **Generation Process**: Beyond the brief mention of Claude, what were the specific prompts or processes used to create the 13,612 instruction-response pairs for the Section, Content, and HTML generators?
* **Quality Validation**: Given its synthetic origin and the "high-quality" claim, what steps (if any) were taken to manually or automatically verify the factual accuracy, coherence, and overall quality of these generated pairs?

---

> ### Author Response · Authors · 2025-11-22
> **Response to Reviewer Fp4t (1/2)**
>
> We sincerely thank you for your detailed and constructive review. Your feedback is extremely valuable. We appreciate the opportunity to provide these clarifications below. We will incorporate all of these details into the final manuscript to significantly improve its reproducibility and readability.
>
> ---
>
> ## Regarding Weakness 1: Lack of Detail on P2P Checker Mechanisms
>
> Thank you for pointing this out. The checker modules are indeed the core of our quality assurance mechanism. We will add a new subsection in the Appendix with the following implementation details and case studies.
>
> *   **Figure Checker:** This module is **rule-based and programmatic**.
>     *   For figure extraction, we use the pre-trained DocLayout-YOLO model, as this sub-task is not the focus of our research.
>     *   The "initial confidence threshold" is set to **0.85**. This high value minimizes false positives and ensures that only high-confidence visual elements are initially extracted.
>     *   The reflection loop is triggered if the count of detected figures mismatches the count of detected captions (which using Manhattan distance to pair them). In such cases, the threshold is **iteratively decreased by 0.05**. This allows Figure Extractor to capture elements that the model may have initially assigned a lower confidence score to (e.g., tables that visually resemble plain text), ensuring all significant visuals are reliably captured without introducing low-confidence noise.
>
> *   **Section Checker:** This module uses an **LLM-as-a-Judge**. After the Content Generator produces the poster's text, this checker prompts LLM to evaluate the output against the source paper based on the four metrics mentioned (coherence, completeness, faithfulness, and correct referencing).
>     *   The LLM returns a binary decision ("OK" or "Problem"). If a problem is detected (e.g., missing content, incorrect figure reference), the LLM also provides specific feedback for revision.
>     *   The reflection loop is then initiated, and the Content Generator re-runs with this new feedback, ensuring the output is refined until it passes the check.
>
> *   **Poster Checker:** This module is **rule-based** for efficiency and accuracy.
>     *   It programmatically parses the generated HTML to check for structural integrity (e.g., no rendering errors) and layout aesthetics. For example, if the **proportion of blank space exceeds 10%**, the reflection loop is triggered. Other layout analysis indicators are listed in Table 5.
>     *   The checker then feeds key layout metrics from the failed attempt (e.g., the high whitespace ratio, imbalanced column sizes) back to the HTML Generator as additional context, instructing it to regenerate the layout until the aesthetic and structural requirements are met.
>
> ---
>
> ## Regarding Weakness 2: Missing Implementation Details for P2P Agents
>
> We agree that these details are crucial for reproducibility. We will expand Appendix F to make our experimental setup perfectly clear.
>
> Aside from the fixed components (`PyMuPDF` for text extraction, `DocLayout-YOLO` for figure extraction, and the programmatic Figure Checker and Poster Checker), all other agents and their sub-components (**Figure Describer, Section Generator, Content Generator, HTML Generator**) are powered by the **same underlying LLM** being evaluated in a given experiment.
>
> For example, when evaluating `GPT-4o`, all of these generative components use the `GPT-4o` API. This setup allows us to fairly benchmark each model's ability on the complex paper-to-poster task, with the results shown in Table 1. As noted by footnote [T], for text-only LLMs(e.g., Deepseek), the visual `Figure Describer` task is handled by `Claude-3.7-Sonnet`.
>
> ---
>
> ## Regarding Weakness 3: Contradictory Description of Fine-Grained Evaluation
>
> We sincerely apologize for the confusing and contradictory wording. We failed to clearly distinguish the two sequential steps of the process. We will revise the paper to clarify the procedure as follows:
>
> The evaluation **methodology is deterministic**, but it **uses an LLM as a tool** for automated verification.
>
> 1.  **LLM as a Verification Tool:** We use an LLM-as-a-Judge (`GPT-4o`) as an automated tool. Its task is not to provide a subjective judgment but to perform a **strictly defined check**. For *each item* on the human-authored checklist, the LLM is prompted to verify whether that specific fact is accurately represented in the generated poster.
>
> 2.  **Deterministic Scoring Methodology:** The final score calculation is **deterministic and is not a trained model**. It programmatically aggregates the verification results from the LLM using the formula provided on L285: `S_fine = (Σ s_i / Σ M_i) * 100`.
>
> In short, the LLM automates the tedious, item-by-item verification process, while a deterministic algorithm aggregates these verification outputs into a final, objective score. To eliminate any ambiguity, we will revise Section 3.2.1 to clearly delineate these two steps.

---

> > ### Author Response · Authors · 2025-11-22
> > **Response to Reviewer Fp4t (2/2)**
> >
> > ## Regarding Question 1: Unclear Notations
> >
> > Thank you for pointing these out. We will clarify them in the manuscript.
> > *   **`v_i` (L134):** This refers to the **raw visual element itself** (the image file cropped from the paper, stored in PNG/base64 format) along with its metadata (bounding box, width, height, aspect ratio). It is not a feature vector.
> > *   **`S` (L142):** This refers to the **structural schema**, which is a **JSON object**. The keys are the poster's section titles, and the values are short textual descriptions of the intended content for that section (e.g., `{"Introduction": "Introduce the research problem..."}`). This is also illustrated in the Content Generator of Figure 1.
> >
> >
> > ## Regarding Question 2: Detail of the `P2Pinstruct` Dataset
> >
> > We apologize for the omission and will add a new Appendix section with the following details.
> >
> > *   **Data Source:** The dataset was constructed using the training split of **SciPostLayout**, which is separate from our test set. We chose this source because its CC-BY license permits derivative works and because it offers topic diversity beyond computer science. We initially considered using papers from ICLR, but after consulting with legal professionals and reviewing the [official ICLR website](https://iclr.cc/FAQ/Copyright), we concluded that the copyright remains with the original authors, and obtaining individual permissions for dataset construction would be infeasible. Our choice reflects a commitment to ethical data sourcing.
> >
> > *   **Generation Process:** The dataset consists of input-output pairs from each stage of the `P2P` framework. To ensure high quality, we guide the generation process by **providing elements of the ground-truth author-created poster as a reference at each step.** For example:
> >     *   The Content Generator receives the paper's text but is also provided with the text extracted from the ground-truth poster for reference.
> >     *   The HTML Generator receives the generated Markdown content but also sees the ground-truth poster's image (PDF converted to PNG) as a visual layout reference.
> >     This process decomposes the creation of a human-made poster into the discrete steps of our `P2P` pipeline, creating high-fidelity instruction pairs.
> >
> > *   **Quality Validation:** The "high-quality" claim is supported by three pillars:
> >     1.  **Built-in Quality Control:** The data was generated within the full `P2P` system with the **checker-reflection loops enabled** and with **guidance from ground-truth examples**, ensuring a high baseline of factual fidelity and structural coherence.
> >     2.  **Manual Verification:** We randomly sampled 20 input-output pairs from each generation stage and manually verified their factual and structural quality, confirming a high level of accuracy.
> >     3.  **Empirical Validation:** As shown in Table 1, models fine-tuned on `P2Pinstruct` consistently and significantly outperform their base versions, providing strong empirical evidence of the dataset's effectiveness and quality.
> >
> > ---
> >
> > Thank you once again for your meticulous and helpful feedback. We are confident that by incorporating these clarifications, the final paper will be substantially stronger and more reproducible. We hope our detailed responses have fully addressed your concerns.

---

> > > ### Author Response · Authors · 2025-11-28
> > > **A Friendly Reminder for Reviewer Fp4t**
> > >
> > > Dear Reviewer Fp4t,
> > >
> > > Thank you again for your thoughtful and constructive comments. We wanted to kindly let you know that we have already provided detailed responses to your questions regarding **the implementation of our checker-reflection mechanisms, the methodology for the Fine-Grained Evaluation, and detail of P2Pinstruct** over the past six days. We have also incorporated these clarifications into the updated version of the paper.
> > >
> > > **We hope our responses and the revised manuscript have thoroughly addressed your questions, and we look forward to any further discussion or feedback.**

---

### Official Review · Reviewer_c6H3 · 2025-11-01

**Soundness:** 3
**Presentation:** 3
**Contribution:** 3
**Rating:** 4
**Confidence:** 3

**Summary:**

This work introduces a technically sound framework for automating academic poster generation, supported by a novel dataset and a well-designed evaluation benchmark. The paper is well-organized and addresses a meaningful problem in academic communication. However, several methodological details require further clarification to ensure the robustness and generalizability of the proposed approach.

**Strengths:**

The paper makes three core contributions:  A multi-agent system that decomposes poster generation into specialized sub-tasks (figure extraction, content summarization, and layout assembly) with integrated reflection mechanisms for iterative improvement.  A large-scale instruction dataset containing over 30,000 examples, designed to support training and fine-tuning of models for the poster generation task.  A benchmark featuring fine-grained checklists and a universal evaluation metric, combining objective fidelity checks with learned subjective quality assessment.

- The modular design of the P2P framework demonstrates a clear understanding of the complexities involved in translating academic papers into visual summaries.
- The release of P2PINSTRUCT and P2PEVAL represents a valuable resource for the community, enabling reproducible research and direct comparison of future methods.
- The dual evaluation strategy (fine-grained and universal) effectively captures both factual accuracy and aesthetic quality, addressing a critical challenge in evaluating generative tasks.

**Weaknesses:**

1. The checker-reflection paradigm represents a significant architectural innovation, but its failure boundaries remain unclear. Could you provide a typology of errors that persist despite reflection cycles? Specifically, we're interested in cases where the system's compositional reasoning breaks down - for instance, when reconciling complex multi-panel figures with nuanced methodological descriptions. Understanding these limitations would help define the theoretical ceiling of this approach.
2. The improvements from P2PINSTRUCT fine-tuning are clear, but we should examine whether this comes at the cost of creative diversity. Are fine-tuned models converging toward a "P2P house style" that prioritizes template adherence over adaptive design? Quantitative analysis of layout diversity and qualitative assessment of creative risk-taking in generated posters would help address concerns about potential over-standardization of academic communication.
3. The recursive nature of P2PINSTRUCT generation raises fundamental questions about knowledge distillation in synthetic datasets. Beyond potential error propagation, we should examine whether this approach creates an "imprinting bias" where the dataset becomes increasingly optimized for the P2P framework's specific architectural assumptions. Could you provide analysis showing how the statistical distribution of generated examples evolves through this process and whether it converges toward a local optimum that might limit future model innovation?
4.  While P2PEVAL's scale is commendable, the selection methodology for the 121 paper-poster pairs warrants deeper scrutiny. More importantly, have you investigated whether the benchmark exhibits structural biases toward certain paper formats (e.g., NLP papers with standardized experimental sections) that might disadvantage models trained on more diverse corpora? The field would benefit from understanding how benchmark composition affects comparative model performance beyond aggregate scores.
5. The choice of XGBoost for universal scoring, while pragmatic, raises questions about the conceptual framework for evaluating creative artifacts. Have you conducted cross-architecture validation to test whether your learned metric captures fundamental design principles versus simply recognizing patterns characteristic of P2P-generated content? The community needs assurance that this evaluation approach generalizes to novel generation paradigms.

**Questions:**

Some suggestions for improvement
- Include statistical significance testing for key comparisons in Table 1 to strengthen claims of improvement.
- Expand the failure analysis section to provide qualitative examples of error correction via reflection loops.
- Discuss the computational overhead of the full P2P pipeline (with reflection) to help users assess practical deployability.

---

> ### Author Response · Authors · 2025-11-24
> **Response to Reviewer c6H3 (1/2)**
>
> We are sincerely grateful to the reviewer for this deeply thoughtful and intellectually engaging review. Your questions have pushed us to clarify the conceptual framework and long-term implications of our contributions. We appreciate the opportunity to address these critical points.
>
> ---
>
> ## W1: Failure Boundaries of the Checker-Reflection Paradigm
>
> This is a critical question about the theoretical ceiling of our approach. Our checker-reflection paradigm is effective at correcting syntactic and structural errors, but its ability to resolve deep semantic or complex compositional reasoning errors is, as you keenly observe, bounded by the capabilities of the underlying LLM.
>
> * **Correctable Errors (Structural/Syntactic)**: The reflection loops excel at fixing issues like general layout imbalances (e.g., excessive whitespace), missing sections, or HTML rendering errors. These problems are reliably detected by programmatic checks or direct LLM verification, and the correction is often straightforward.
> * **Persistent Errors (Complex Compositional Reasoning)**: The system struggles when the underlying LLM has a fundamental misunderstanding of complex, domain-specific content. For instance, if a paper contains numerous, intricate multi-panel figures that require a highly specific arrangement to be coherent, the LLM might struggle to generate a logical HTML structure. The *Poster Checker* may detect the resulting layout imbalance (e.g., high whitespace), but if the root cause is the LLM's inability to reason about the complex spatial relationship between sub-figures, reflection alone may not solve it. The LLM, lacking a better strategy, might simply reshuffle the same flawed components.
> * **Our Mitigation Strategy**: In these cases, we employ an escalation strategy. If a checker fails after a set number of retries (e.g., 5), we escalate the reflection loop to an **earlier agent**. For example, a persistent layout failure in the *Orchestrate Agent* can trigger the *Section Agent* to regenerate its content entirely, attempting to provide a simpler or more structured input that the *Orchestrate Agent* can handle. This provides a multi-level recovery mechanism, though it is ultimately still bounded by the LLM's core reasoning ability.
>
> We will add a new appendix section with illustrated examples.
>
> ---
>
> ## W2: Creative Diversity vs. a "P2P House Style"
>
> This is an important concern regarding the impact of instruction tuning. We argue that fine-tuning on `P2Pinstruct` does not lead to a monolithic "house style," but rather teaches models to apply **effective design patterns**, which enhances creative success instead of stifling it.
> *  **`P2P` is Inherently Adaptive, Not Template-Based:** Unlike prior work [1] that relies on fixed templates, our framework is designed for adaptive design. The multi-agent approach dynamically generates a structure and layout tailored to each paper's unique content. Our `P2Peval` benchmark also penalizes rigid, formulaic layouts (like those from YuanBao), rewarding adaptive design instead.
> *  **Learning Principles, Not Memorizing Styles:** The fine-tuned models do not converge to a single style. Instead, they learn to avoid common pitfalls (like cluttered layouts or poor flow) by mastering a vocabulary of effective design principles. The significant performance gains (e.g., our fine-tuned 8B model outperforming the much larger GPT-4.1-mini) demonstrate that the models are learning generalizable skills, not just mimicking a template. The diverse visual examples in our appendix further provide qualitative evidence of this layout diversity.
>
> [1] PostDoc: Generating Poster from a Long Multimodal Document Using Deep Submodular Optimization. Vijay Jaisankar,  et al. 2024.
>
> ---
>
> ## W3: "Imprinting Bias" in `P2Pinstruct`
> This is a fundamental question about the synthetic datasets. We mitigate the risk of "imprinting bias" and convergence to a local optimum by **anchoring our data generation process to external human ground truth.**
> Our process is not a simple recursive loop. At each stage of generation, the model is guided by the corresponding part of the original, human-created poster. This **guided synthesis** or "teacher-forcing-like" approach ensures the dataset learns to map the steps of a human design process to our P2P architecture, rather than falling into a self-referential echo chamber. For example,
>    * The *Content Generator* is prompted to summarize the paper while using text from the ground-truth poster as a reference.
>    * The *HTML Generator* receives the generated Markdown but also sees the ground-truth poster's image as a visual layout reference.
>
> This methodology is a key contribution for building robust instruction sets for complex, multi-stage tasks. The consistent improvements shown in Table 1, across both lexical and quality metrics, serve as empirical evidence that the dataset captures useful, generalizable knowledge, not just an overfit to our specific architecture.

---

> ### Author Response · Authors · 2025-11-24
> **Response to Reviewer c6H3 (2/2)**
>
> ## W4: Structural Bias in the `P2Peval`
> Benchmark fairness was a central design principle for `P2Peval`. We took deliberate steps to ensure diversity and avoid bias towards specific paper formats.
> 1.  **Diverse by Design:** As detailed in Appendix C and shown in Figure 4, the papers are sourced from both CS conferences and the multi-disciplinary SciPostLayout (covering Biology, Medicine, etc.). This was a conscious choice to ensure the P2Peval tests general capabilities.
> 2.  **Performance Breakdown as Evidence Against Bias:** The results in Tables 7 & 8 provide the strongest evidence against structural bias. For instance, best LLM achieves its highest Fine-Grained score in *Psychology* and its highest Universal score in *Computer Science*. If the benchmark were biased toward a standard "NLP paper" format, we would expect CS papers to dominate all metrics. The fact that different domains excel on different quality axes indicates that `P2Peval` is well-balanced and effectively measures LLM's ability to adapt across disciplines and styles.
>
> ---
>
> ## W5: Generalizability of the XGBoost-based Universal Score
> This is a crucial question about the long-term validity of our evaluation metric. `P2Peval` was a principled decision to ensure the metric captures fundamental design principles rather than surface-level patterns, thereby ensuring its generalizability.
> The key to this is two-fold:
> 1.  **Diverse Data:** The XGBoost was trained on ratings of posters generated by a highly diverse set, ranging from top-tier closed models to small open-source ones. The training data was therefore not a uniform "P2P style" but a wide distribution of quality and design paradigms. The model learned to align with human preference across this entire spectrum.
> 2.  **Principled Features:** The model's inputs (U1–U10) are not arbitrary features; they are based on universal principles of graphic design and scientific communication(e.g., balance, fidelity).
> Because the model learns how humans weigh these universal principles when viewing a wide variety of poster styles, it is far more likely to generalize to novel generation methods than a model trained on raw pixels or unprincipled features. The study in our response to Reviewer HwQu further validates that this hybrid approach is more aligned with human judgment (R²=0.92) than direct end-to-end LLM scoring (R²=0.27).
>
> ---
>
> ## Suggestions for Improvement
>
> Thank you for these excellent suggestions. To avoid cluttering the rebuttal, we will include the full analyses in our **upcoming rebuttal revision PDF**. Below is a summary of key findings.
>
> *   **Statistical Significance:** We agree this is crucial for strengthening our claims. Following best practices from statistical research [2, 3, 4], we report **Effect Size (Cohen's d)** instead of p-values, as it better quantifies the practical magnitude and significance of the observed differences. We use the standard convention where |d|>0.8 indicates a Large effect, |d|>0.5 a Medium effect, and |d|>0.2 a Small effect.
>
>     | Effect Size of Fine-Tuning on `P2Pinstruct` | FineGrain Cohen's d | Universal Cohen's d |
>     | - | - | - |
>     |InternVL3-8B|2.13|1.72|
>     |Qwen2.5-VL-7B|4.18|3.00|
>     |Qwen3-8B|1.21|0.60|
>
>     | Effect Size of Ablation Study (vs. Full `P2P`) | FineGrain Cohen's d | Universal Cohen's d |
>     | - | - | - |
>     |Remove Reflection|-0.15|-0.61|
>     |Remove Figure Describer|-0.26|-0.43|
>     |Only Multi-Agent|-0.28|-0.82|
>     |End-to-End|-0.72|-0.60|
>
>     Fine-tuning on `P2Pinstruct` yields a **Large Effect Size** across all models for the Fine-Grained metric and at least a Medium Effect Size for the Universal metric. This confirms that `P2Pinstruct` provides meaningful improvements. And this analysis further validates that removing key components results in a **Medium-to-Large Negative effect** on quality.
>
> *   **Failure Analysis:** As discussed in response to W1, the rebuttal PDF will feature a new appendix section.
>
> *   **Computational Overhead:** We agree this is vital for users. We will add a detailed cost and latency analysis. To summarize, generating a poster for an average paper costs approximately 0.25 dollars with GPT-4.1 and only 0.02 dollars with Qwen-2.5-VL-7B, demonstrating the `P2P`'s practical affordability and tunable cost-performance trade-off.
>
> [2] The ASA Statement on p-Values: Context, Process, and Purpose. Ronald L. Wasserstein, et al. 2016.
>
> [3] Effectiveness of Transdiagnostic Cognitive-Behavioral Psychotherapy Compared With Management as Usual for Youth With Common Mental Health Problems: A Randomized Clinical Trial. Pia Jeppesen, et al. 2021.
>
> [4] The sertraline vs. electrical current therapy for treating depression clinical study: results from a factorial, randomized, controlled trial. Andre R Brunoni, et al. 2013.
>
> ---
>
> We thank you again for deep engagement with our work. We are confident that incorporating these clarifications will make our paper substantially stronger.

---

> > ### Author Response · Authors · 2025-11-28
> > **A Friendly Reminder for Reviewer c6H3**
> >
> > Dear Reviewer c6H3,
> >
> > Thank you again for your thoughtful and constructive comments. We wanted to kindly let you know that we have already provided detailed responses to your questions regarding **the theoretical boundaries of checker-reflection paradigm, the potential for "imprinting bias" in P2Pinstruct, and the generalizability of evaluation metrics** over the past four days. We have also incorporated these conceptual discussions and new statistical analyses into the updated version of the paper.
> >
> > **We hope our responses and the revised manuscript have thoroughly addressed your questions, and we look forward to any further discussion or feedback.**

---

### Author Response · Authors · 2025-11-26
**Author Response and Summary of Revision**

We would like to express our sincere gratitude to Reviewers c6H3, Fp4t, HwQu and 4iW7 for their thorough, insightful, and constructive feedback. Your detailed reviews have been invaluable in helping us strengthen and clarify our work.

Several clarifications and new analyses have been incorporated into the revised manuscript, directly addressing the points raised during the review period. Key updates include:

*   **Clarified the Checker-Reflection Mechanism (W1/c6H3, W1/Fp4t, Q3/HwQu, Q3/4iW7):** We have added a new **Appendix B and Figure 5** with detailed implementation specifics for each checker, including failure triggers, reflection examples, and a discussion on the framework's compositional reasoning boundaries and escalation strategies.

*   **Resolved Ambiguity in Fine-Grained Evaluation (W3/Fp4t, W2/HwQu):** We have rewritten **Section 3.2.1** to explicitly describe the two-step process: (1) using an LLM for automated verification against a human-authored checklist, followed by (2) a deterministic scoring algorithm. This clarifies that the metric measures content fidelity, not layout imitation.

*   **Justified the Universal Score Methodology (W5/c6H3, W3/HwQu, Q2/4iW7):** We added **Table 9** , rewrote **Appendix F.3 and Section 3.2.2** with a new ablation study that empirically validates our XGBoost-based approach. The results show it is significantly more aligned with human judgment (R²=0.92) than simpler, end-to-end LLM scoring alternatives (R²=0.27).

*   **Expanded on the Details of `P2Pinstruct` Dataset (W3/c6H3, Q2/Fp4t, W1/HwQu):** A new **Appendix D** now details the dataset's teacher-forcing-like generation process, which anchors it to human-created posters to mitigate imprinting bias. It also outlines our quality validation pillars.

*   **Included Statistical Significance(Q1/c6H3):** As requested, we included **Appendix H** with statistical significance testing (Cohen's d) for our main results.

*   **Added Cost Analysis (Q3/c6H3, W4/4iW7):** We added **Appendix K** with a detailed breakdown of the computational cost and latency for our framework, demonstrating its practical affordability.

*   **Strengthened Discussion on Limitations(W1/c6H3, W3/4iW7):** We added a new **Limitations section** to the main paper discussing output formats and reasoning boundaries.

We believe these revisions make the paper substantially stronger, clearer, and more reproducible. We thank the reviewers once again for their deep engagement and valuable guidance.

**We hope our individual responses and the revised manuscript have thoroughly addressed your questions, and we look forward to any further discussion or feedback.**

---

### Author Response · Authors · 2025-12-03
**Summary of Rebuttal (1/2)**

Dear AC,

Thank you very much for your time and for overseeing the review process of our paper.
Due to the OpenReview bug that interrupted the discussion cycle, we would like to provide a consolidated summary to help you quickly see (i) what reviewers appreciated, and (ii) how we have addressed their main concerns in the revised manuscript.
Overall, the reviewers recognized our work as a significant and comprehensive contribution to an underexplored problem, but raised important questions regarding methodological clarity, benchmark validity, and practical considerations. We believe our revisions and new analyses have fully addressed these points, substantially strengthening the paper.

---

## Strengths Highlighted by the Reviewers

All four reviewers acknowledged the novelty and value of our main contributions, establishing a strong foundation for the work:

*   **A Comprehensive Ecosystem:** All reviewers highlighted the impressive scope of our work as a complete, end-to-end solution, from the generation framework (`P2P`) to the training data (`P2Pinstruct`) and evaluation benchmark (`P2Peval`).
*   **Innovative `P2P` Framework:** The multi-agent architecture with its `checker-reflection` loop was consistently praised as a novel and effective design for a complex document transformation task.
*   **Valuable Community Resources:** The release of `P2Pinstruct` (the first large-scale instruction dataset for this task) and `P2Peval` (a comprehensive benchmark with 1738 checklist items) was seen as a major contribution that will enable reproducible research.
*   **Principled Dual-Evaluation in `P2Peval`:** The benchmark's core innovation (decoupling objective fidelity from subjective quality) was recognized as a well-motivated approach to evaluating creative AI artifacts.
*   **Experimental thoroughness**: The evaluation on 35 models (closed and open source), the ablation study on P2P components, the human preference study, and analysis of output formats (HTML vs. LaTeX/SVG) are consistently noted as strong points. Reviewers find the experimental results convincing and the system effective.


---

## Concerns Raised by Reviewers & Our Brief Responses

The reviewers’ main concerns were about methodological clarity and potential biases. Below, we summarize each concern and our response.

### 1. Checker–Reflection Mechanism: Ambiguity in Implementation and Limits
*(c6H3, Fp4t, HwQu, 4iW7)*

**Concern:** The checkers, reflection loops, and their effectiveness were not clearly defined.

**Our Response:** We added **Appendix B and Figure 5** with concrete implementation details. We clarified that the Figure and Poster Checkers are rule-based, while the Section Checker uses an LLM. We also detailed failure triggers, escalation strategies, and analyzed the system’s error-correction boundaries.


### 2. Fine-Grained Evaluation: Confusing Methodology
*(Fp4t, HwQu)*

**Concern:** Apparent contradiction between “LLM-as-a-Judge” and “deterministic methodology”; what exactly is being measured.

**Our Response:** We rewrote **Section 3.2.1** to clarify the two-step process: (1) using an LLM for automated verification against a human-authored checklist, followed by (2) a deterministic scoring algorithm. We now state explicitly that the metric measures **content fidelity, not layout imitation**.

### 3. Universal Evaluation: Justification and Generalizability
*(c6H3, HwQu, 4iW7)*

**Concern:** The two-step LLM-featurizer + XGBoost pipeline for the Universal score was not justified over simpler, direct LLM judging.

**Our Response:** We conducted a new study (**Table 9, Appendix F.3**) empirically validating our approach. The results show our method is significantly more aligned with human judgment (R²=0.92) than end-to-end LLM scoring (R²=0.27), justifying the design.

---

> ### Author Response · Authors · 2025-12-03
> **Summary of Rebuttal (2/2)**
>
> ### 4. `P2PINSTRUCT` Dataset: Potential for Bias and Lack of Detail
> *(c6H3, Fp4t, HwQu, 4iW7)*
>
> **Concern:** The synthetic dataset might inherit and amplify biases from the P2P framework itself, creating a "closed loop".
>
> **Our Response:** We clarified in **Appendix D** that `P2Pinstruct` is generated via a “teacher-forcing-like” process, where each generation step is anchored to a ground-truth, human-created poster. This mitigates self-referential bias. We also added statistical analysis (**Appendix H**) showing that fine-tuning on `P2Pinstruct` yields significant and large effect sizes on both Fine-Grained and Universal metrics.
>
> ### 5. `P2PEVAL` Benchmark: Potential for Structural Bias
> *(c6H3, 4iW7)*
>
> **Concern:** The benchmark might be biased toward certain paper formats (e.g., standard CS/NLP papers).
>
> **Our Response:** We now highlight in **Appendix E.3** and **Tables 7 & 8** that the benchmark was sourced from diverse fields (CS, Biology, Medicine, etc.). The per-domain performance analysis shows that different fields excel on different metrics, providing evidence against structural bias.
>
> ### 6. Practical Limitations: Output Formats and Cost
> *(4iW7, HwQu)*
>
> **Concern:** The paper did not sufficiently discuss the limitation of using HTML or the system’s computational cost.
>
> **Our Response:** We moved this discussion into a new **Limitations section** in the main paper and added a detailed cost/latency breakdown in **Appendix K**, demonstrating the practical affordability and tunable trade-offs of our framework.
>
> ---
>
> We thank the reviewers for their invaluable guidance. We are confident that these extensive revisions and analyses have addressed all concerns raised by the reviewers. The paper now presents a clearer, more robust, and more thoroughly validated ecosystem for automated poster generation.
>
> Thank you once again for your time and dedication to the ICLR review process.

---

### Meta-Review · Area_Chair_VN4Y · 2026-01-07

**Summary:**

Revs. had an extended list of very relevant questions and clarifications. Here I report only the most important ones for my final evaluation.
1. Rev. c6H3, Fp4t and 4iW7 asked for more analysis, details, and use cases of the checker-reflection approach.
2. Rev. c6H3, HwQu and 4iW7 asked for more details and motivation about the MLLM-Featurized + XBoost-based scoring.
3. Rev. Fp4t, HwQu, 4iW7 asked for more details, how it is built, possible risks and how effective P2Pinstruct dataset is to train a model to learn to generate posters from paper descriptions.
4. Rev. HwQu asked for clarifications about the distinction between fidelity and imitation in their fine-grained scoring.
4. Rev. 4iW7 asked for a comparison with non-LMM-based models.
5. Rev. 4iW7 is concerned about the limitation of the proposed method to only HTML-based posters.

**Reviewer Concerns:**

1. The checker-reflection approach is one important contribution of the proposed P2P LLM-based agent. In the rebuttal, the authors effectively provide a clearer understanding of how the approach works, accompanied by some simple use cases.
2. The authors explained the rationale behind the XBoost-based scoring and showed in a new experiment that the proposed evaluation is better than direct LLM judging or other approaches.
3. The authors provided more details about the synthetic dataset, explained how it was built and how its quality was manually verified. Finally, the gains in terms of performance of a base model compared with a model fine-tuned on the dataset showed its effectiveness.
4. The authors could not really compare with previous non-LLM-based approaches as they could not find code available.
5. Effectively, the proposed approach can work only with HTML-based posters, which is a clear limitation of the approach.

The authors provided a strong rebuttal answering in an honest and clear way to most of the questions raised by the authors. Although some concerns cannot be solved (e.g. comparison with non-LLM approaches or limitation to HTML-based posters), overall, the rebuttal helped to clarify and better understand the contributions of the paper.

**Reviewer Scores:**

- Rev. c6H3: 4 -> 6
Rev. concerns were more about clarification and a better understanding of certain points. The authors did a good job in clarifying and adding important information about their contribution. Therefore, I believe that Rev. c6H3 would have increased their form 4 score to 6.
- Rev. Fp4t: 6 -> 6
Rev. asked mostly for clarifications and details. The authors provided all the necessary information, therefore, rev. maintained their score.
- Rev. HwQu 4 -> 6
Rev. asked for some additional analysis (on the P2Pinstruct) and a discussion about imitation and fidelity.  The authors' answers were clear and easy to follow. Thus, I think rev. would have increased the score to 6.
- Rev. 4iW7 6 -> 6
Rev. asked many important questions and most of them were clearly answered. The limitation of the proposed approach to only HTML poster generation could be a reason why rev. would not increase their score.

---

### Decision · Program_Chairs · 2026-01-26

Accept (Poster)